# Multimodal Prescriptive Deep Learning

## Abstract

We introduce a multimodal deep learning framework, Prescriptive Neural Networks (PNNs), that combines ideas from optimization and machine learning, and is, to the best of our knowledge, the first prescriptive method to handle multimodal data. The PNN is a feedforward neural network trained on embeddings to output an outcome-optimizing prescription. In two real-world multimodal datasets, we demonstrate that PNNs prescribe treatments that are able to significantly improve estimated outcomes in transcatheter aortic valve replacement (TAVR) procedures by reducing estimated postoperative complication rates by over 40% and in liver trauma injuries by reducing estimated mortality rates by 25%. In four real-world, unimodal tabular datasets, we demonstrate that PNNs outperform or perform comparably to other well-known, state-of-the-art prescriptive models; importantly, on tabular datasets, we also recover interpretability through knowledge distillation, fitting interpretable Optimal Classification Tree models onto the PNN prescriptions as classification targets, which is critical for many real-world applications. Finally, we demonstrate that our multimodal PNN models achieve stability across randomized data splits comparable to other prescriptive methods and produce realistic prescriptions across the different datasets.

## 1 Introduction

Today's society provides an increasing availability of large quantities of data, particularly multimodal data consisting of structured and unstructured elements. As a result, developing systematic and personalized decision-making methods that can leverage such multimodal data becomes more and more critical, and the benefits of data-driven methods become more and more visible. For example, medical professionals could systematically and optimally treat patients based on individual characteristics, clinical notes, and medical scans (Soenksen et al., 2022). Companies in technology and digital advertising would be able to increase customer impact by customizing content and advertisements according to user-specific data. In the retail industry, such personalized models would allow companies to dynamically price goods and services based on the user or environment for increased revenue.

Much of the current work in machine learning and deep learning focuses on improving the accuracy of output prediction. We find that deep learning has tremendous and underutilized potential in the area of decision-making. This paper combines ideas from machine learning and optimization to move from prediction to prescription, with the ability to leverage multimodal data. We introduce a novel, multimodal, deep learning framework that we call a Prescriptive Neural Network (PNN). We demonstrate how our models handle complex data structures and how effective they are in both multimodal and unimodal real-world applications. Through these applications, we show that our PNN models are flexible with different treatment scenarios that cover all real-life application settings. Our models also give stable and realistic results, comparable to existing prescriptive methods, and provide the user with more control over the resulting prescriptions. On tabular datasets in particular, we are able to recover interpretability by applying knowledge distillation and fitting interpretable Optimal Classification Tree (OCT) models (Bertsimas & Dunn, 2017; 2019) on the PNNs' prescriptions as classification targets; we find that these Mirrored OCTs perform comparably to their PNN counterparts, meaning that interpretability comes with little cost to performance.

## 1.1 Related Literature

Previous literature in data-driven personalized decision-making includes the Regress & Compare framework, tree-based methods, and causal methods.

**Regress & Compare.** Regress & Compare is a black-box methodology where a regression model is trained to predict the outcome under each treatment. The set of features used for training is the augmented feature data combined with the historical treatment given. Given an input observation and possible treatment options, the model is then used to select, for each sample, the treatment with the lowest (highest) outcome for a minimization (maximization) problem.

In this area, Zhao et al. (2012) introduces a framework that aims to estimate individual treatment rules. The goal is to assign treatments that maximize (or minimize) the expected outcome for each individual, by estimating the potential outcomes under each treatment (using an outcome weighted learning approach) and selecting the treatment that leads to the best outcome.

There are also many applications of the Regress & Compare methodology for prediction – examples include energy economics (Ferkingstad et al., 2011) and multidrug-resistant tuberculosis (Siddique et al., 2019). Other works use the Regress & Compare framework to move from predictions to prescriptions. In particular, Bertsimas & Kallus (2020) extends Regress & Compare solutions for prescriptive problems and incorporates $k$-nearest neighbors regression ($k$NN Altman (1992)), local linear regression (LOESS Cleveland & Devlin (1988)), classification and regression trees (CART Breiman et al. (1984)), and random forests (RF Breiman (2001)). Although Bertsimas & Kallus (2020) demonstrates that their methods are widely applicable and computationally tractable under mild conditions, we note that these are classical machine learning methods and do not take advantage of deep learning.

More specific applications of Regress & Compare for prescriptive problems include healthcare (Bertsimas et al., 2017a; Bayati et al., 2014) and revenue management (Bertsimas & Kallus, 2022). Bertsimas et al. (2017a) considers personalized diabetes treatment, while Bayati et al. (2014) combines prediction and decision-making to allocate interventions for post-discharge patients that were admitted due to heart failure. Bertsimas & Kallus (2022) considers the problem of optimal pricing, where they learn from historical observational data to optimize predicted revenue given price.

One possible limitation of the Regress & Compare approach is that it is affected by the number of samples per treatment, since for each treatment, only the samples that received that treatment in real life are considered. Also, it does not address the potential treatment assignment bias present in the data; e.g. healthier patients tend to receive lighter treatment and to have better outcomes. This is discussed in more detail in Section 2.3. Furthermore, the black-box nature reduces its interpretability, which is important for many real-world applications.

**Tree-based methods.** Kallus (2017) introduces Personalization Trees, which extend the Regress & Compare method for the problem of choosing the treatment with the best causal effect from a finite number of discrete options. Kallus presents three different recursive-partitioning-based algorithms: a greedy Personalization Tree, a Personalization Forest that bags Personalization Trees, and a globally optimal Personalization Tree. As these are tree methods, we note that they preserve interpretability.

Bertsimas et al. (2019) introduces Optimal Prescriptive Trees, which are similar to Personalization Trees but combine the counterfactual estimation and prescriptive learning tasks together in one training process and extend the framework of Optimal Classification Trees from Bertsimas & Dunn (2017; 2019). As a result, the trees are highly interpretable. Amram et al. (2022) further explores the optimal trees methodology and proposes Optimal Policy Trees, in which counterfactual estimation is performed separately from the prescriptive learning task. This allows for greater flexibility in discrete and continuous treatments, as well as better learning of the prescriptive task due to reduced complexity that results from the separation of the two training tasks. Like Optimal Prescriptive Trees, this method also preserves interpretability. These approaches, however, struggle with learning more complicated functional forms and are therefore limited to learning outcome functions that can be modeled by trees of relatively small depth.

**Causal methods.** This family of methods originates from the causal inference literature and includes both individual trees (causal trees) and their combinations (causal forests). Athey & Imbens (2016) introduces causal trees, which employ a recursive partitioning approach of the feature space to split the data into groups with similar treatment effects. Causal forests extend causal trees and represent a prescriptive black-box method that builds on random forests, as introduced by Wager & Athey (2018). While random forests are constructed from decision trees, causal forests are composed of causal trees, which aim to maximize the difference in outcomes between two treatments at each node during tree growth. The resulting outcomes are interpreted relatively to one another. In the binary treatment case, since there are only two options (treatment or no treatment), one option will yield a positive effect (outcome) and the other a negative effect. If the goal is to minimize the outcome, the treatment option with the negative effect is prescribed.

Other models in the causal inference literature include causal boosting (Powers et al., 2018) and causal MARS (Powers et al., 2018). However, the estimation of treatment effects, which is achieved by causal models, is not an explicit policy prescription, which is the goal of this work.

Finally, another approach by Zhou et al. (2023) takes inspiration from the causal inference literature and uses inverse propensity weight estimators to calculate the counterfactuals. Decision trees (both greedy and fully optimal) are then used for policy learning. Fully optimal trees, however, struggle with scalability, while the heuristic-based trees do not guarantee the best possible policy (optimality).

**Deep learning methods.** Others have taken a deep learning approach to the optimal prescription problem. Patil et al. (2024) introduces prescriptive networks that are shallow neural networks to address the binary treatment regime, in which a treatment may or may not be given. Their networks are optimized by over-estimating conditional average treatment effects (CATE), and they propose a method using mixed-integer programming (MIP) to implement their networks into commercial solvers. Sun & Tsiourvas (2023) proposes a piecewise linear neural network model to output optimal prescriptions from a set of discrete treatments and show that their model partitions the input space into disjoint polyhedra, where all observations in the same partition are assigned the same treatment. Bergman et al. (2022) proposes a solver that takes as input user-specified pretrained predictive models (including neural networks) and formulates optimization models directly over those predictive models to provide final prescriptions.

Additionally, Shalit et al. (2017); Shi et al. (2019) lie at the intersection of causal methods and deep learning; they use neural networks to estimate causal effects. We briefly note that this is different from directly prescribing treatments to solve the optimal prescription problem. Shalit et al. (2017) proposes a general framework called Counterfactual Regression (CFR) and its variant, the Treatment-Agnostic Representation Network (TARNet). These models are designed to facilitate Individual Treatment Effect (ITE) estimation through a fully differentiable learning process that employs a regularized objective optimized via a deep feed-forward network consisting of six exponential-linear activation layers. Similarly, Shi et al. (2019) introduces Dragonnet, an alternative neural architecture tailored for ITE estimation. Its architecture is a three-headed structure that jointly models the propensity score and potential outcomes.

We note that, like our PNN models, all of these works combine ideas from optimization and machine learning. However, Bergman et al. (2022) does not incorporate Deep Learning in the prescriptive part of the framework, but only to generate predictions. Patil et al. (2024); Sun & Tsiourvas (2023) consider binary and discrete treatments respectively, while our work handles more treatment and outcome scenarios. Furthermore, our approach differs in the network's objective function used for training. Finally, Shalit et al. (2017); Shi et al. (2019) are methods for causal estimation rather than direct prescription purposes, and though they is a possibility for extension to discrete treatments, they both only consider and report results for binary treatments.

## 1.2 Contributions

Our contributions are as follows:

1. Combining machine learning and optimization, we propose a novel, multimodal, deep learning framework we call a Prescriptive Neural Network (PNN); to the best of our knowledge, our model is the first prescriptive method to handle multimodal data.

2. In two real-world multimodal datasets, we demonstrate that PNNs prescribe treatments that are able to significantly improve estimated outcomes in transcatheter aortic valve replacement (TAVR) procedures by reducing estimated postoperative complication rates by over 40% and in liver trauma injuries by reducing estimated mortality rates by 25%. Additionally, PNNs either outperform or perform comparably to existing, state-of-the-art, prescriptive methods on four real-world unimodal (tabular) datasets that span all four treatment scenarios: diabetes management (multiple continuous treatments), groceries pricing (single continuous treatment), splenic injuries treatment (multiple discrete treatments), and REBOAs in blunt trauma patients (binary treatment).

3. On tabular datasets we recover interpretability through knowledge distillation; we train Optimal Classification Trees (OCT) (Bertsimas & Dunn, 2017; 2019) on the feature data but using the PNN prescriptions as target classes, similar to a binary or multiclass classification task. We call these Mirrored OCTs. Remarkably, the performance of the Mirrored OCTs is equally strong as that of the original PNNs, with a decrease in improvement of only 3.38% on average across the tabular datasets; this implies that interpretability may be recovered with minimal cost to performance.

4. Finally, we demonstrate that our multimodal PNN models achieve stability across randomized data splits comparable to other prescriptive methods and produce realistic prescriptions across the different datasets.

## 2 Methods

In this section, we review the methodology of our PNNs. We first formally define the prescriptive problem we seek to solve (Section 2.1), and then we present the training process, which is divided into four main steps: embedding extraction (Section 2.2), counterfactual estimation (Section 2.3), prescription policy learning (Section 2.4), and interpretability recovery (Section 2.5).

### 2.1 Problem definition

Formally, we consider a prescription problem, which can be characterized by observational data in the form $\{(\boldsymbol{x}_i, y_i, t_i)\}_{i=1}^n$:

- **Features** $\boldsymbol{x}_i \in \mathbb{R}^p$ is the $p$-dimensional feature data for the $i$-th observation.

- **Treatment** $t_i \in \mathcal{T}$ is the treatment applied historically to the $i$-th observation, where $\mathcal{T}$ is the set of all possible treatments. As treatments may be discrete or continuous, there are four possible treatment scenarios: binary (treatment or no treatment), multiple discrete (two or more treatment options), single continuous (one treatment option with continuous values), or multiple continuous (two or more treatment options, some or all taking on continuous values).

- **Outcome** $y_i \in \mathbb{R}$ is the result observed after treatment $t_i \in \mathcal{T}$ has been applied to the $i$-th observation.

Given this observational data, the aim is to develop an optimal prescriptive model that outputs a treatment $t \in \mathcal{T}$ that results in an optimal outcome $y$ for each input observation $\boldsymbol{x}$.

### 2.2 Embedding Extraction

The first step in the model pipeline is to extract embeddings from structured and unstructured data.

#### 2.2.1 Structured data

We extract embeddings from structured feature data using traditional pre-processing techniques as described below, where the technique depends on whether the feature is numerical, categorical, or ordinal.

- **Numerical features.** Numerical features are normalized to the interval [0,1] by subtracting the minimum feature value and dividing by the feature range; we do this to increase stability and equal weighting of features during counterfactual estimation and model. We note that since tree models are independent of data scale, we use the original feature values when training all tree models, which ensure interpretability in the tree splits.

- **Categorical features.** For categorical features, we use one-hot encodings to convert them to binary features, such that each category becomes a new indicator feature.

- **Ordinal features.** Ordinal features are categorical features whose values carry numerical information. Since these categories have a natural order to them, we can assign each category a number such as 1 to 5, where relative magnitude holds information. The feature value assigned to the number "1" conveys that that value is less than that of a feature value assigned the number "4." These ordinal features are then treated as numerical features in our experiments.

### 2.2.2 Unstructured data

We extract embeddings from unstructured data using pretrained, deep learning models. By passing each observation's unstructured data through these pretrained models, we can obtain a vector representation of the unstructured datapoint. In particular, our experiments on medical data in Section 3 use Clinical Longformer (Li et al., 2022), a long sequence transformer model trained via a sparse attention mechanism on domain-specific, large-scale clinical corpora; from this model, we obtain a 768-dimensional embedding vector for each observation's text data.

An important aspect of the multimodal component lies in handling the extracted embeddings. When the dimensionality of these embeddings is high relative to the dataset size, it can lead to overfitting and training instability (Advani et al., 2020). To mitigate this, we explore dimensionality reduction techniques such as Principal Component Analysis (PCA), as well as extracting a compact representation from an intermediate layer of a classification head fine-tuned on the outcome. Though optional, this step can improve training stability, tractability, and downstream performance in our PNN model. In particular, we reduce the clinical note embeddings to 32 dimensions in our medical experiments, though this number can be adjusted depending on the application and dataset size. For the PCA-based reduction, the selected dimensions retain more than 95% of the original variance in all datasets considered.

While we specifically use ClinicalLongformer, any pretrained large language model (LLM) may be used to process unstructured text data. Additionally, any pretrained computer vision (CV) model may be used to process unstructured image data. This results in an embedding extraction step for unstructured data that is not only highly accessible, but also highly flexible.

To get the final multimodal embeddings, we concatenate the individual modalities' embeddings to obtain one large embedding vector.

### 2.3 Counterfactual Estimation

The next step is counterfactual estimation. Because the prescriptive problems' dataset only contains historical observational data, the counterfactuals are unknown, e.g. the hypothetical outcomes $y(\boldsymbol{x}_i, t)$ for $t \neq t_i$, for each observation $\boldsymbol{x}_i$. We therefore perform a counterfactual estimation step (Dudik et al., 2011) that estimates the outcomes for each observation under every treatment. This produces a rewards matrix $\Gamma$, where $\Gamma_{i,t}$ is the estimated outcome of applying treatment $t$ to the $i^{\text{th}}$ observation. The estimation process is slightly different for discrete and continuous treatments.

### 2.3.1 Counterfactual estimation for discrete treatments

We use two methods for counterfactual estimation of discrete treatments. The doubly-robust method is, however, preferred for almost all of the experiments in Section 3, since it addresses the treatment assignment bias. The two methods are as follows:

1. **Direct Method.** This method directly learns the outcome function $y_t(\boldsymbol{x})$ by training separate models, one for each treatment $t$. During training, each model uses only the subset of the observations that received treatment $t$. These models can be random forests or boosting methods and output an estimated outcome $\hat{y}_t(\boldsymbol{x})$ for when treatment $t$ is hypothetically applied to observation $\boldsymbol{x}$.

2. **Doubly robust estimation.** Because direct estimation is often prone to treatment assignment bias, the doubly-robust estimator attempts to mitigate this bias by re-weighting the estimated direct outcomes with propensity score probabilities. This reweighting is expressed in Equation (1), which calculates the doubly-robust reward matrix $\Gamma$:

$$\Gamma_{i,t} = \hat{y}_{i,t} + \mathbb{1}\{t_i = t\}\frac{1}{p_{i,t}}(y_i - \hat{y}_{i,t}), \tag{1}$$

where $\hat{y}_{i,t} = \hat{y}_t(\boldsymbol{x}_i)$ is the estimated outcome of sample $i$ under treatment $t$, $p_{i,t} = \mathbb{P}[t_i = t]$ is the probability that treatment $t$ is assigned to observation $i$ in real life and $y_i$ is the actual outcome of observation $i$. To reduce the potential instability that arises when we divide with the probability $p_{i,t}$, we clip the ones that are smaller than a certain value (Lee et al., 2011). We generally choose a clipping threshold of 0.01-0.05, depending on the resulting rewards' values.

For binary outcomes, we use classifiers for counterfactual estimation, while for continuous outcomes, we use regressors.

### 2.3.2 Counterfactual estimation of continuous treatments

For continuous treatments, we train a regression model to predict the outcome of the $i^{\text{th}}$ observation using as input the observational data $\boldsymbol{x}_i$ and continuous prescribed treatment doses $T_{i,t}$ for each treatment $t$. Then, by discretizing the continuous dose values and only considering a subset of them as valid treatments, we use the trained model to retrieve the estimated outcome for the $i^{\text{th}}$ observation under all valid treatment schemes. This is analogous to most real-world treatment scenarios; when we handle continuous treatments, we always select a subset of the possible ones, since the real-world treatment options need to be finite.

### 2.4 Prescription policy learning through feedforward neural networks

At its core, the architecture of our Prescriptive Neural Network (PNN) is the classical feedforward neural network trained via backpropagation (Rosenblatt (1958); Rumelhart et al. (1986)). Without loss of generality, we assume our goal is to minimize outcomes in the prescriptive problem. The objective of our prescriptive neural network is to minimize total rewards for the prescriptions $\tau(\boldsymbol{x}_i)$ assigned by the network to each observation $\boldsymbol{x}_i$ in the dataset:

$$\min_{\tau(.)} \sum_{i=1}^{n} \sum_{t\in\mathcal{T}} \mathbb{1}\{\tau(\boldsymbol{x}_i) = t\} \cdot \Gamma_{i,t}, \tag{2}$$

proposed by Amram et al. (2022). Because the indicator function is not differentiable, the backpropagation algorithm cannot handle Equation (2) exactly. We therefore "soften" the objective and leverage an approach analogous to that of multi-classification networks. The PNN assigns treatments probabilistically, such that its output layer consists of $|\mathcal{T}|$ neurons, one for each distinct treatment (as multi-classification networks have an output corresponding to each target class). We denote the output vector of the PNN as $\boldsymbol{z} \in \mathbb{R}^{|\mathcal{T}|}$ and apply a softmax activation function to these output neurons to obtain a probability distribution over the distinct treatments.

We obtain the final prescription of the network by finding the treatment $t$ with the highest probability $\mathbb{P}[\tau(\boldsymbol{x}_i) = t] = \sigma_t(\boldsymbol{z})$. This approach is analogous to a classification network, where the predicted class is the one with the highest probability among the network's output nodes. The tractable objective for our PNN models is therefore:

$$\min_{\tau(.)} \frac{1}{n} \sum_{i=1}^{n} \sum_{t\in\mathcal{T}} \mathbb{P}[\tau(\boldsymbol{x}_i) = t] \cdot \Gamma_{i,t}. \tag{3}$$

### 2.4.1 Convergence properties

The loss function (3) shares key properties with the cross-entropy loss function commonly used in multiclass classification problems. The cross-entropy loss is defined as:

$$L(\theta) = -\frac{1}{n} \sum_{i=1}^{N} \sum_{t \in \mathcal{T}} y_{i,t} \sigma_t(\boldsymbol{z}), \tag{4}$$

where $y_{i,t}$ is the true label of sample $i$, and $\sigma_t(\boldsymbol{z})$ is the softmax probability for class $t$. The cross-entropy loss is widely used because it is smooth, has bounded gradients, and is Lipschitz continuous. These properties contribute to the efficient convergence of optimization algorithms like SGD and Adam (Kingma & Ba, 2014; Bottou, 2010).

Similarly, the loss function (3) employed in this work exhibits these desirable properties. Specifically:

- Smoothness: The loss function is smooth because it is a linear combination of the softmax probabilities $\sigma_t(\boldsymbol{z})$, which are themselves smooth functions (Bishop & Nasrabadi, 2006).

- Bounded Gradients: The gradients of the loss function are bounded, since the weights $\Gamma_{it}$ are bounded due to clipping (as described in Section 2.3.1) and the derivative of the softmax function is also bounded (Goodfellow et al., 2016).

- Lipschitz Continuity: The loss function is Lipschitz continuous because the softmax function is Lipschitz continuous, and the weights $\Gamma_{it}$ are bounded (Nesterov, 2013).

The primary difference between our loss function and the cross-entropy loss is that our loss function uses weights $\Gamma_{i,t}$ instead of true labels $y_{i,t}$, and it does not include the logarithm of the probabilities. However, these differences do not fundamentally alter the smoothness, boundedness, or Lipschitz continuity of the loss function. As a result, the convergence behavior of our loss function is similar to that of the cross-entropy loss when training feedforward neural networks for multiclass classification problems (LeCun et al., 2015).

Under the assumption of bounded weights in the network, a property typically observed when training with SGD or Adam (Ghadimi & Lan, 2013; Reddi et al., 2019), the network will converge to critical points of the loss function. This is consistent with the behavior observed in standard neural network training with cross-entropy loss (Zhang et al., 2016).

### 2.5 Recovering Interpretability with Optimal Classification Trees

For structured datasets, we are able to recover interpretability through the use of knowledge distillation, in which we fit Optimal Classification Trees (OCTs) (Bertsimas & Dunn, 2017; 2019) on the feature data and prescription outputs of the PNN. We present an example of such a Mirrored OCT in Figure 1 (with other examples available in Appendix A.7). This example comes from the REBOA (resuscitative endovascular balloon occlusion of the aorta) application in Section 3.5. In this real-world problem, we aim to minimize patient mortality by either prescribing (treatment 1) or not prescribing (treatment 0) the REBOA treatment. The tree in the figure is fit on the same observational data used to train its corresponding PNN, while the PNN prescriptions are used as target classes. We observe that this tree is very interpretable, and the features chosen for the splits come from our structured, observational data. If a patient (sample) is assigned to a leaf where the prediction is 0, then they are not prescribed the treatment, and if they are assigned to a leaf where the prediction is 1, the REBOA treatment is recommended.

## 3 Experiments with real-world datasets

In this section, we apply PNNs on real-world datasets that are both multimodal and unimodal. We first review methodology for data splits, network architecture, and performance evaluation, which are relevant for all of our experiments. We then report results and relevant discussions for each of our two multimodal datasets and four unimodal datasets.

Figure 1: Example of REBOA Mirrored OCT.

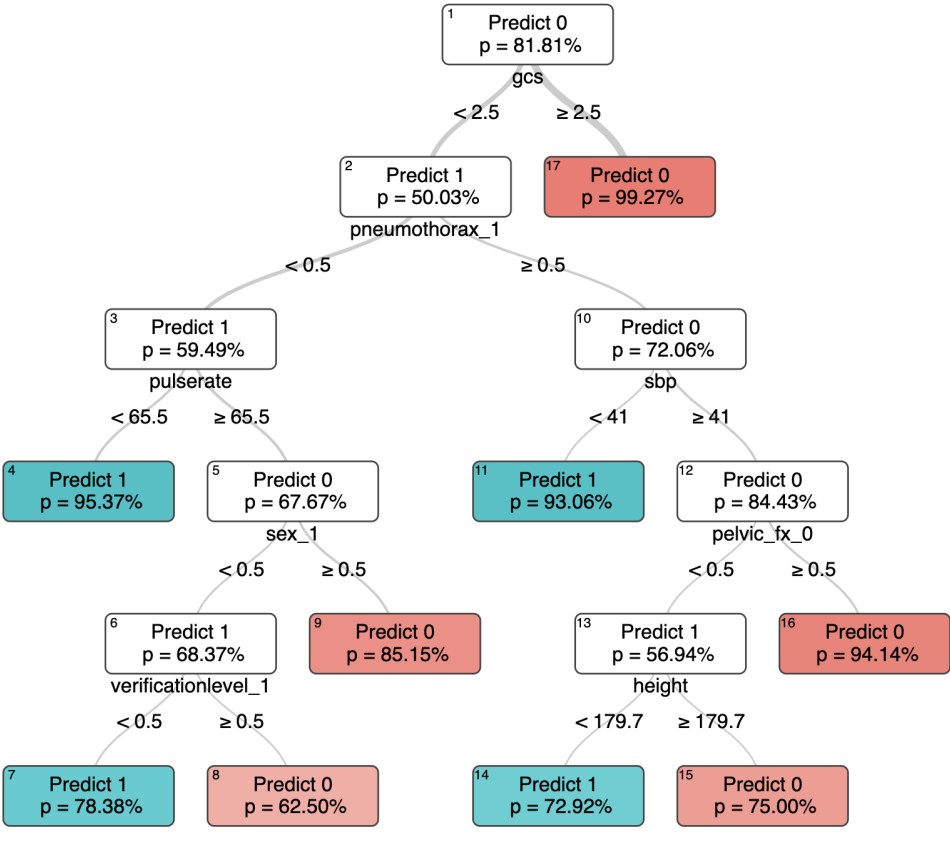

## 3.1 Data splits

We split each dataset evenly, using 50% for the training and 50% for the test set. This choice is attributed to the fact that in order to evaluate the performance of the prescriptive methods on a test set, knowing the outcomes of the samples under the different treatments is required. Furthermore, to prevent data leakage between training and test splits and unfair performance, the counterfactual estimation for the test set is necessary and performed separately from that of the training set. Because we need enough data points to ensure high-quality counterfactual estimation of both the training and test sets, the typical ratios of 80-20 or 70-30 are not applicable in this case.

## 3.2 Network Architecture

We uniquely tune the PNN architecture for each dataset. We specify and finetune the hyperparameters of the PNN in Appendix A.2. In addition to these architectural choices, we choose the Adam optimizer. We tune the aforementioned hyperparameters using a validation set we extract from the training set that is not used in model training. We typically keep 15% of the training data for the validation set. The Mirrored OCTs are then trained on the prescriptions from each of the PNN models.

## 3.3 Experiments

To evaluate the models' performance on each dataset, we perform multiple train-validation-test splits and report the average performance of each split's models. This ensures that the results are not tailored to a specific data split, and also enables the investigation of stability of the prescriptive methods. Also, given the

randomness often associated with training machine learning models, we train multiple models per split and also average their performance. In total, we perform 5 randomized data splits per dataset, and we train 5 models per split, so we train 25 models in total, for each model type.

## 3.4 Performance metrics

To assess model performance, we use a relative outcome improvement metric measured on the unseen test set. This metric compares the estimated outcome of the treatment prescribed by the model with that of the real-life treatment. For most datasets, where doubly robust estimation is applied, test-set reward matrix entries lack natural meaning. Thus, instead of comparing with the actual outcome and to ensure fairness, hypothetical outcomes for both treatments (model-prescribed and real-life) are drawn from the reward matrix and then compared. The average relative outcome improvement is then computed as:

$$\bar{I} = \frac{\sum_{i=1}^{n} |\Gamma_{i,\hat{t}_i} - \Gamma_{i,t_i}|}{\sum_{i=1}^{n} \Gamma_{i,t_i}}, \tag{5}$$

where $\Gamma_{i,\hat{t}_i}$ is the estimated outcome for the $i^{\text{th}}$ observation if the prescribed treatment $\hat{t}_i \in \mathcal{T}$ is applied and $\Gamma_{i,t_i}$ is the estimated outcome for the $i^{\text{th}}$ observation under the real-life treatment $t_i$.

For the case of unstructured datasets, the evaluation on the test set can be performed using counterfactuals that have been calculated either by using a single modality, or multimodal data. To address that, the outcome improvement is evaluated using the test set counterfactuals with both types of models (in this case, tabular, and multimodal, from tabular and notes).

## 3.5 Unstructured datasets

We demonstrate the efficacy of our PNN models on two real-world, multimodal datasets: transcatheter aortic valve replcement (TAVR) and liver trauma injuries. Both datasets include tabular (structured) and clinical notes (unstructured) data, and we train two models: unimodal models – fit on just the tabular data – and multimodal models – fit on the combined tabular and notes data. We report the results of both datasets and discuss the improved performance of the multimodal pipeline in this section.

**Transcatheter aortic valve replacement (TAVR).** Transcatheter aortic valve replacement (TAVR) is a treatment option for patients with severe aortic stenosis across all levels of surgical risk. In the United States, two transcatheter heart valves (THV) are used, the balloon-expandable Edwards Sapien 3 and the self-expanding Medtronic Evolut Pro Plus. Selection of valve choice by medical professionals is generally based on several factors including operator preference, patient characteristics, and valvular/annular anatomy on a computerized tomography (CT) scan (Mitsis et al., 2022; Leone et al., 2023). Despite improvement in TAVR devices, implant techniques, and operator experience, permanent pacemaker implantation (PPI) continues to remain a frequent complication with an estimated prevalence of 7-18% (Webb & Wood, 2012; Smith et al., 2011), with potential consequences on patients' mortality and cost of care. This dataset contains demographic (e.g. age, sex, bmi) and medical information (e.g. hypertension, Left Ventricular Ejection Fraction), as well as radiology reports of echocardiograms and CT scans from 2,127 patients, and the problem we seek to solve is prescribe the most appropriate type of valve to patients, so that their risk of PPI is minimized. We train two different sets of models, one where only the tabular features are considered, and one where notes are also incorporated, in the form of embeddings, extracted as described in Section 2.2.2.

**Liver trauma injuries.** Acute liver injury is considered one of the two most common solid organ injuries in blunt trauma victims. However, inaccuracies exist in the grading of liver injuries by human read and interpretation of CT scans, which may lead to mistreatment (Georg et al., 2014). Therefore, personalized treatment for the patient is important in trauma management. This dataset comes from electronic medical records of 722 liver injury patients and includes features such as patient demographics, history of illness, lab results, and allergies. We aim to prescribe either surgical or non-surgical intervention to minimize patient mortality (binary outcome).

In both datasets, to assess the impact of multimodal augmentation and to ensure there is no bias stemming from which modality is used to perform the reward estimation in the test set, we train separate reward models on the tabular and multimodal datasets and evaluate all models using both modalities' estimated rewards for a fair comparison. In the multimodal case, rewards are always estimated using the full-dimensional embeddings from the pretrained language model, both for training and evaluation, to preserve the full information for the rewards. We then train PNNs using three variants of the text embeddings: the full embeddings, a reduced representation from a fine-tuned classification head, and PCA-reduced embeddings. We report performance using the models trained on the type of embeddings that yield the best results for each dataset on the validation set; namely the classification head embeddings for the TAVR dataset and the PCA-reduced embeddings for the liver injury dataset. The final improvements are reported in Table 1. Dimensionality reduction is overall beneficial, as models trained on full embeddings are less stable (see Appendix A.1).

For both of the datasets and particularly in the TAVR dataset, we observe that the multimodal models generally outperform the unimodal ones, demonstrating the benefit of increased information from the added language modality. Although the discrepancy between the outcome improvement can be quite different depending on the modality used to train the test set reward estimators, we observe a pattern of improvement when multimodality is employed under both estimators. The results are also stable, across 5 different data splits and 5 different models per split, and indicate the prescriptive power multimodality can offer.

Table 1: Improvement(%) in risk for the experiments with unstructured data (TAVR, liver trauma), where lower risk is better. We report the average improvement and standard error across the five splits.

| Estimator | Method | TAVR models | | Liver trauma models | |
|---|---|---|---|---|---|
| | | Tabular | Multimodal | Tabular | Multimodal |
| Tabular | PNN | $5.05 \pm 2.60$ | $17.87 \pm 6.24$ | $14.85 \pm 4.39$ | $21.74 \pm 1.96$ |
| | Mirrored OCT | $7.67 \pm 3.45$ | $17.14 \pm 7.41$ | $26.77 \pm 1.61$ | $26.46 \pm 1.77$ |
| Tabular | PNN | $21.09 \pm 1.08$ | $\mathbf{42.89 \pm 5.64}$ | $23.14 \pm 1.66$ | $25.25 \pm 3.00$ |
| & Notes | Mirrored OCT | $\mathbf{22.58 \pm 2.50}$ | $41.66 \pm 7.00$ | $\mathbf{29.14 \pm 2.27}$ | $\mathbf{29.15 \pm 2.17}$ |

Table 2: Training accuracy (%) of the Mirrored OCTs for the unstructured datasets. We report the average accuracy and standard error across the five splits.

| Dataset | Tabular model | Multimodal model |
|---|---|---|
| TAVR | $79.06 \pm 2.23$ | $92.47 \pm 3.14$ |
| Liver trauma | $86.77 \pm 0.78$ | $85.30 \pm 0.85$ |

Mirrored OCTs, trained on the PNNs' prescriptions, result in an outcome improvement comparable or even better to the PNNs, in both datasets, demonstrating that Mirrored OCTs do not generally result in performance decrease. We also quantify the approximation error between the PNN and OCT prediction using the training accuracy of the OCTs on the prescriptions, which we present in Table 2. For example, a training accuracy of 79% means that in the training data split, the OCT correctly classified 79% of the PNN's prescriptions. We note that both of the liver dataset's OCTs' training accuracies are lower than in other unstructured and structured datasets, which likely contributes to a smaller gap between the respective tabular and multimodal OCT performances in that dataset.

## 3.6 Structured datasets

We now apply our PNN models to four real-world, unimodal tabular datasets: diabetes management, groceries pricing, splenic injuries treatment, and REBOA in blunt trauma patients. Because these are purely tabular datasets, we are able to recover interpretability by fitting Mirrored OCT models. For more details

on the treatment scenarios covered by these four datasets, the counterfactual estimation method employed, please refer to Table 13 in Appendix A.3.

Table 3: Improvement(%) for structured datasets. We report the average improvement and standard error across the five splits.

| Method | Diabetes | Groceries | Spleen | REBOA |
|---|---|---|---|---|
| Regress & Compare | $1.74 \pm 0.21$ | $94.17 \pm 6.25$ | $7.11 \pm 2.30$ | $-10.60 \pm 8.85$ |
| Causal Forest | $0.43 \pm 0.16$ | $98.68 \pm 2.67$ | $5.68 \pm 4.56$ | $-20.77 \pm 8.91$ |
| Optimal Policy Tree | $1.36 \pm 0.26$ | $106.58 \pm 2.38$ | $12.26 \pm 2.85$ | $7.28 \pm 2.57$ |
| PNN | $\mathbf{1.83 \pm 0.23}$ | $\mathbf{110.88 \pm 1.18}$ | $13.50 \pm 2.55$ | $\mathbf{8.22 \pm 1.74}$ |
| Mirrored OCT | $1.75 \pm 0.26$ | $110.22 \pm 3.10$ | $\mathbf{13.83 \pm 1.90}$ | $7.41 \pm 2.3$ |

Table 4: Training accuracy (%) of the Mirrored OCTs for the structured datasets. We report the average accuracy and standard error across the five splits.

| Diabetes | Groceries | Spleen | REBOA |
|---|---|---|---|
| $92.06 \pm 1.18$ | $98.03.47 \pm 0.81$ | $94.03 \pm 1.59$ | $97.12 \pm 0.34$ |

We present results for all four structured datasets in Table 3, where we directly compare PNNs and their Mirrored OCTs with the performance of other well-known, state-of-the-art prescriptive methods, including Optimal Policy Trees, Regress & Compare, and Causal Forests. We note that for Regress & Compare, we typically train an XGBoost Regressor or Classifier (depending on the nature of the outcome). For this purpose, we append the actual treatment as a separate column in the observational data and we train the predictive model to predict the real-life outcome under the treatment. To select the best treatment for a new sample, we append each of the available treatments separately and we obtain the final outcome in each case. The treatment that results in the best outcome is selected. We report the training accuracy of the Mirrored OCTs for these structured datasets in Table 4.

**Diabetes management.** This dataset is based on electronic medical records of 58,200 patients with type 2 diabetes from 1999 to 2014 from the Boston Medical Center. It contains information regarding patient demographics, a timeseries of insulin levels, as well as current drug prescriptions. Patient treatments include combinations of insulin, metformin, and oral blood glucose regulation agents, and patient outputs are the resulting hemoglobin A1C measurements (continuous outcome), for which lower values are more optimal.

**Groceries pricing.** For this study, we select the publicly-available retail dataset "The Complete Journey" (Lugauer et al., 2020; Biggs et al., 2021), which contains household-level transactions of many products over two years of 2,500 frequent-shopper households. We focus on one specific product, strawberries. The task here is to, given household demographics, prescribe optimal prices to strawberries with a binary outcome indicating if the household purchases strawberries or not after being assigned the strawberry price. The objective is to maximize revenue, where revenue is defined as the price if strawberries are purchased and zero otherwise (e.g., binary). After filtering the data to only the relevant households that had purchased strawberries at least once, the final dataset consists of 97,295 transactions. We impute strawberry prices for cases where strawberry-purchasing households did not purchase strawberries on that specific trip by using the mode of the strawberry prices on the most recent day prior to the trip on which no strawberries were purchased. We consider prices from $2 to $5, inclusive, in increments of $0.50. Since there does not seem to be a strong correlation between strawberry price and the covariate features, rewards are estimated using the direct method.

**Splenic injuries treatment.** The spleen is an immunologic intra-abdominal organ on the left side of the body, which may be removed in the case of injury. In the 1970's to 1980's, the medical community saw a shift towards preservation of the spleen rather than removal, thus making it important to correctly determine if spleen removal was indeed necessary. This specific dataset includes data on spleen surgical

operations, in addition to demographic and medical data consisting of numerical, binary, and categorical types. After preprocessing, we have 35,954 rows of patient data in this dataset. We aim to optimally prescribe splenectomy, angioembolization, or observation in blunt splenic injuries to minimize patient mortality (binary outcome).

**REBOA in blunt trauma patients.** The use of resuscitative endovascular balloon occlusion of the aorta (REBOA) for control of noncompressible torso hemorrhage continues to be highly debated. Being able to appropriately determine if such a treatment should be used is critical in order to decrease the misuse of the treatment in hemodynamically unstable blunt trauma patients. This dataset includes 9,998 patients, with features that are both demographic and medical in nature, including numerical, binary, and categorical values. The goal is to prescribe the REBOA treatment or not to minimize patient mortality (binary outcome). Some feature columns contain unknown values; we therefore use Optimal Imputation (Bertsimas et al., 2017b) with K-Nearest Neighbors to fill the missing values. A few features are integral, and we round imputed values to the nearest integer to maintain integrality.

As shown in Table 3[1], we observe consistent improvements in estimated outcomes across all structured datasets and methods. PNNs and Mirrored OCTs generally achieve the strongest results across tasks, with PNNs achieving the highest average improvement in three of the four datasets (Diabetes, Groceries, and REBOA), and Mirrored OCTs performing best in the Spleen dataset.

The strong and consistent performance of PNNs across datasets highlights their robustness, with results that are statistically comparable or better than other methods. Beyond performance, a key strength of PNNs lies in their flexibility: the same architecture is directly applicable to multimodal datasets. This makes PNNs particularly appealing in settings where both structured and unstructured data must be handled jointly within a unified prescriptive framework.

## 4 Discussion

### 4.1 Relevant causal inference topics

Before applying the prescriptive model, we examine the suitability of each dataset from a causal inference perspective. We begin by evaluating the positivity assumption, complemented by a treatment support diagnostic (effective treatment count). We then establish that the doubly robust method we use for counterfactual estimation is both necessary and effective in addressing covariate imbalance across treatment groups.

#### 4.1.1 Positivity

To ensure the validity of causal estimates, we first examine the positivity assumption — the requirement that each individual has a non-zero probability of receiving all treatments. This condition is essential for valid estimation of causal effects, as counterfactual inference is not possible in regions where certain treatments were never assigned (Petersen et al., 2012).

To evaluate overlap, we calculate the propensity score distributions $P(t \mid \boldsymbol{x})$ for each treatment group and visualize them using Gaussian kernel density estimates (KDEs). We then compute a numeric average overlap (AO) score based on the area of intersection between KDEs for all treatment pairs and all treatments as targets.

For a given treatment $t \in \mathcal{T}$, and any two treatment groups $t_i, t_j$, we define the pairwise overlap:

$$O_{t_i,t_j}^{(t)} = \int_0^1 \min\left(f_{t_i}^{(t)}(s), f_{t_j}^{(t)}(s)\right) ds, \tag{6}$$

and compute the average across treatment pairs and targets:

---

[1]For the groceries dataset, improvement is computed as mean revenue improvement rather than outcome improvement, where mean revenue is $\bar{pr} = \frac{1}{n} \sum i = 1^n \Gamma_{i,\hat{t}i} \cdot \hat{t}i$, and actual revenue is $\bar{pr} = \frac{1}{n} \sum i = 1^n y_{i,t_i} \cdot t_i$, with $\hat{t}i$ the prescribed treatment for sample $i$ by the model, and $ti$ the real-life treatment.

$$\text{AO} = \frac{1}{|\mathcal{T}| \cdot \binom{|\mathcal{T}|}{2}} \sum_{t \in \mathcal{T}} \sum_{t_i < t_j} O_{t_i, t_j}^{(t)}. \tag{7}$$

This score ranges from 0 (no overlap) to 1 (perfect overlap), and provides a quantitative proxy for the strength of positivity in the dataset.

### 4.1.2 Effective Treatment Count

A low overlap score may result from extreme class imbalance — where one or more treatment groups are rarely assigned — rather than lack of support in the covariate space. To account for this, we compute the effective treatment count via the exponential of the Shannon entropy, following Jost (2006):

$$n_{\text{eff}} = \exp\left(-\sum_{t \in \mathcal{T}} p_t \log p_t\right), \tag{8}$$

where $p_t = \frac{n_t}{n}$ is the empirical frequency of treatment $t$. We report the normalization of the effective count by the total number of treatments.

Together, AO and $n_{\text{eff}}$ offer complementary views of treatment positivity: the former captures distributional overlap, while the latter reflects support across treatment arms. Both are shown in Table 5.

Table 5: Average overlap (AO) across propensity distributions and effective treatment counts. For multimodal datasets, we use tabular features only.

| Metric | TAVR | Liver trauma | Diabetes | Groceries | Spleen | REBOA |
|---|---|---|---|---|---|---|
| AO | 0.4494 | 0.5673 | 0.4901 | 0.963 | 0.5841 | 0.0479 |
| Normalized $n_{\text{eff}}$ (%) | 98.5 | 91.0 | 76.3 | 89.4 | 57.0 | 52.5 |

All datasets exhibit AO scores above 0.4, suggesting adequate positivity, with the exception of REBOA. In that case, the overlap score is exceptionally low due to extreme imbalance: only a small number of individuals received the treatment, resulting in a sharply peaked KDE and low intersection. Nevertheless, we include REBOA in our analysis because it represents a realistic and high-stakes medical scenario. To mitigate risk of overinterpretation, we interpret REBOA results cautiously.

The KDEs used to compute AO are shown in Appendix A.4, and the empirical treatment proportions in Figure 9.

### 4.1.3 SMD

We further assess covariate imbalance using the Standardized Mean Difference (SMD) (Rosenbaum & Rubin, 1985), defined for each covariate $\boldsymbol{x}_j, j = 1, \ldots p$ and treatment pair $(t, t_0)$ as:

$$\text{SMD}_i^{(t, t_0)} = \frac{\left|\mu_j^{(t)} - \mu_j^{(t_0)}\right|}{\sqrt{\frac{1}{2}\left(\sigma_j^{2(t)} + \sigma_j^{2(t_0)}\right)}}, \tag{9}$$

where $\mu_j^{(t)}$ and $\sigma_j^2(t)$ denote the mean and variance of covariate $\boldsymbol{x}_j$ within treatment group $t$.

We compute SMDs before and after weighting by the inverse of the estimated propensity scores, $\frac{1}{p_{it}}, \forall i = 1, \ldots n$, and visualize their distributions for each dataset in Figure 2. While the doubly robust estimator does not rely exclusively on weights, we observe that the IPW component alone substantially improves balance: after reweighting, SMDs shift toward 0, and tail density decreases, indicating improved covariate

comparability across groups. For the groceries dataset, we apply the direct method only, as covariate balance is already sufficient without reweighting.

Figure 2: Distribution of SMDs across datasets. Comparisons are relative to a reference group in each dataset. Treatment label keys are in the Appendix Section A.5.

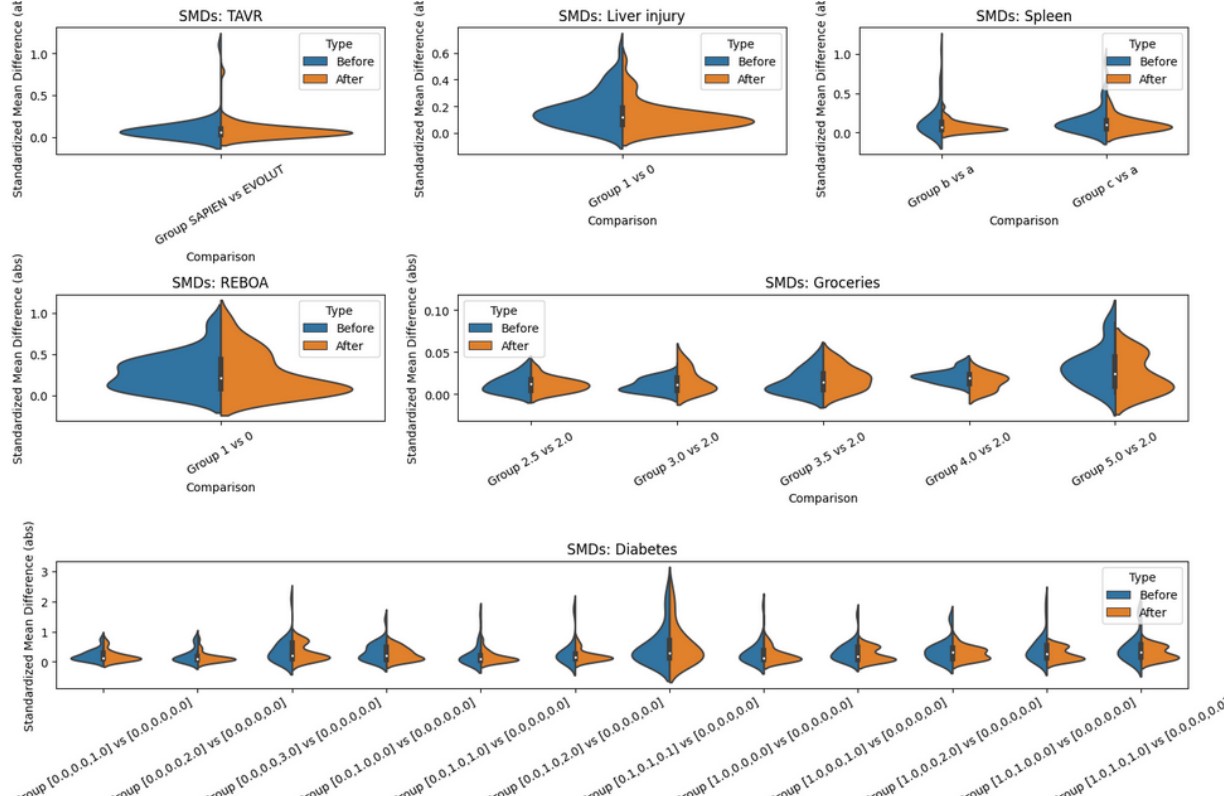

## 4.2 Prescriptions

In general, reporting improvement based on estimated rewards is a good approximation for evaluating the performance of prescriptive methods. However, such metrics do not provide any insights into how realistic the prescriptions are or how different they are from policies observed in the data. Another critical aspect is the models' stability. The prescriptive models should be robust in their prescriptions across dataset splits and with respect to inherent randomness during training. Furthermore, interpretability is crucial for real-world deployment, as users of models must understand where decisions are coming from in order to implement them. Finally, we discuss briefly the potential and challenges for real-world deployment of PNNs and Mirrored OCTs. We therefore discuss these four topics – realistic and stable prescriptions, interpretability, as well as real-world deployment – in the following sections.

### 4.2.1 Realistic Nature of Prescriptions

Providing realistic prescriptions is crucial, particularly when employing prescriptive tools in practice. To evaluate the realism of the provided prescriptions, we quantify the deviation between the prescribed and real-life treatments of individual samples. This evaluation is carried out per model, by calculating the mean absolute difference between each sample's prescribed and real-life treatment throughout the dataset (training, validation, and test sets) and then averaging it across all samples. For discrete cases (REBOA and spleen datasets), the treatments are ordered in terms of severity, so that the distance is reasonable as a metric. For

this purpose, the $N_m = 25$ trained models from each dataset are considered. The mean absolute difference for the $k$-th individual model is given by:

$$D_k = \frac{1}{n} \sum_{i=1}^{n} \|\hat{t}_i - t_i\|_1, \tag{10}$$

where $n$ is the size of the dataset, $\hat{t}_i$ is the prescribed treatment for sample $i$, and $t_i$ is the treatment sample $i$ got in real life. The mean absolute difference across the $N_m = 25$ models is then computed as:

$$\bar{D} = \frac{1}{N_m} \sum_{k=1}^{N_m} D_k. \tag{11}$$

The results are presented in Tables 6 and 7. Clearly, we prefer both high performance $\bar{I}$ and low mean absolute difference $\bar{D}$, as this ensures that improvements in outcomes are not achieved through disproportionate shifts in treatment assignments. We naturally expect that PNN-prescribed treatments are more different than those in real-life, as compared to the other prescriptive methods, since as presented in Sections 3.5 and 3.6, PNNs outperform the other methods in most datasets. Contrary to our expectation, however, we favorably observe that PNNs, as well as the Mirrored OCTs, result in mean absolute difference between the prescribed and the actual treatments that is comparable to the rest of the methods.

Table 6: Mean Absolute Difference between prescribed and actual treatments for structured datasets.

| Method | Diabetes | Groceries | Spleen | REBOA |
|---|---|---|---|---|
| Regress & Compare | **0.4556** | 1.033 | 0.3665 | 0.2815 |
| Causal Forest | 0.5923 | **1.0042** | 0.7761 | 0.2492 |
| Optimal Policy Tree | 0.5329 | 1.0221 | 0.3933 | 0.1608 |
| PNN | 0.5007 | 1.4386 | **0.3658** | 0.1621 |
| Mirrored OCT | 0.4964 | 1.4391 | 0.3685 | **0.1538** |

Table 7: Mean Absolute Difference between prescribed and actual treatments for unstructured datasets.

| Method | TAVR models | | Liver trauma models | |
| | Tabular | Multimodal | Tabular | Multimodal |
|---|---|---|---|---|
| PNN | 0.4085 | **0.7841** | 0.3604 | 0.2863 |
| Mirrored OCT | **0.4082** | 0.5616 | **0.3296** | **0.2847** |

For unstructured data, Section 3.5 highlights the significant advantage of the multimodal approach over tabular-only methods. As expected, this performance edge indicates that the proportion of prescription changes is higher in the multimodal case. The results indicate that, in the TAVR case for example, around 78% of the PNN prescriptions change from one valve to the other in the multimodal case, and 41% in the tabular case, which is a considerable shift. For the liver trauma dataset, the difference between the prescribed and the actual treatments is smaller.

A critical advantage of neural networks, however, is that the user has some control over how much the prescriptions change. For example, depending on the application, a threshold can be selected that limits the number of treatment assignment modifications, and only models that satisfy this constraint on the validation set are considered. Alternatively, one can incorporate a penalty term in the objective function, to penalize an excessive number of treatment switches or to account for different treatment constraints. This is application-specific, but highlights the flexibility that neural networks offer compared to other prescriptive methods.

The realism of prescriptions is also evaluated by examining the average number of distinct prescriptions per type of model, which shows how much the model is capable of utilizing the full treatment space. This is evaluated as:

$$\bar{N} = \frac{1}{|\mathcal{T}|} \cdot \frac{1}{N_m} \sum_{i=1}^{N_m} |t : \exists j : \hat{t}_j = t, \ j = 1, \dots, n|, \tag{12}$$

where the average number of prescriptions is normalized by the size of treatment space, to calculate a percentage and thus make the metric comparable across the different datasets. The results are presented in Tables 8 and 9.

Table 8: Percentage of different prescriptions selected by the models for structured datasets.

| Method | Diabetes | Groceries | Spleen | REBOA |
|---|---|---|---|---|
| Regress & Compare | 36.92 | 23.33 | 66.67 | 70.0 |
| Causal Forest | **99.69** | **95.33** | **88.00** | **100.0** |
| Optimal Policy Tree | 37.54 | 48.67 | 70.67 | **100.0** |
| PNN | 21.23 | 49.33 | 65.33 | **100.0** |
| Mirrored OCT | 21.23 | 49.33 | 64.0 | **100.0** |

Table 9: Percentage of different prescriptions selected by the models for unstructured datasets.

| Method | TAVR models | | Liver trauma models | |
| | Tabular | Multimodal | Tabular | Multimodal |
|---|---|---|---|---|
| PNN | **98.0** | **100.0** | **100.0** | **100.0** |
| Mirrored OCT | 92.0 | 84.0 | 84.0 | 88.0 |

In most of the structured datasets, we observe that PNNs and Mirrored OCTs prescribe a high percentage of the available treatments, with the exception of the diabetes dataset, where the treatment options are multiple. The Regress & Compare approach mostly underutilizes the treatment space, since it prescribes a small percentage of treatments in most cases; this reveals potential treatment assignment bias that may not be mitigated through this approach. Causal Forests seem to make prescriptions that mostly cover the full treatment regime; however their performance is worse than Optimal Policy Trees and PNNs, as discussed in Section 3.6. In the unstructured case, both tabular and multimodal PNN models always employ all of the available treatments. We observe that some of the Mirrored OCTs only prescribed one treatment in the TAVR and liver injury dataset, but most of them prescribed both.

Most importantly, PNNs are flexible in this feature too; by using the dropout mechanism (Srivastava et al., 2014) in the last layers of the network, which is often used to prevent overfitting in neural networks, all of the output nodes are forced to be activated during training. As a result, more areas of the network are used, which empirically shows an increase in selected treatments by the model. The flexibility of PNNs is also underlined by the fact that they provide, for each observation, a probability of each treatment, similarly to a classification problem, where neural networks provide a probability of each class. The results presented for PNNs consider the prescription with the highest probability for each sample. However, one can employ a treatment-specific probability threshold to select the final treatment, like in classification problems; for example, this may be done according to some predefined, meaningful, treatment allocation percentage. This provides the user with some control over the resulting treatment distribution.

Overall, PNNs achieve a balance between performance and realism in prescriptions, and they also reasonably cover the treatment space. These are important factors that make them reliable for practitioners and their leverage in different real-life applications.

### 4.2.2 Stability

Given the randomness that is present when training neural networks, their stability compared to other machine learning models is often criticized (Colbrook et al., 2022). The goal of this section is to compare the stability between the different prescriptive approaches by measuring the standard deviation of each observation's treatment distribution, which results from the $N_m = 25$ different models that have been trained for each dataset. Ideally, the prescriptions should be consistent across the different model runs and data splits; otherwise the method is very sensitive to the training data distribution, which reduces the credibility of the prescriptions.

For each observation, the standard deviation of its prescriptions across the different $N_m$ models is calculated, and we present averages across the different observations in Tables 10[2] and 11.

Table 10: Standard deviation of each sample's prescriptions distribution across $N_m = 25$ models for structured datasets.

| Method | Diabetes | Groceries | Spleen | REBOA |
|---|---|---|---|---|
| Regress & Compare | **0.0752** | **0.0285** | **0.0940** | 0.4230 |
| Causal Forest | 0.5139 | 0.1667 | 0.4535 | 0.3838 |
| Optimal Policy Tree | 0.2654 | 0.7227 | 0.2722 | 0.1436 |
| PNN | 0.3292 | 1.2874 | 0.3001 | 0.2589 |
| Mirrored OCT | 0.1990 | 0.9453 | 0.1882 | **0.0983** |

Table 11: Standard deviation of each sample's prescriptions distribution across $N_m = 25$ models for unstructured datasets.

| Method | TAVR models | | Liver trauma models | |
|---|---|---|---|---|
| | Tabular | Multimodal | Tabular | Multimodal |
| PNN | **0.4232** | **0.3803** | 0.3649 | 0.3260 |
| Mirrored OCT | 0.4512 | 0.4900 | **0.2506** | **0.2185** |

We observe that the standard deviation of PNNs' prescriptions is comparable to the other models across all datasets, which indicates that although training neural networks is associated with inherently more randomness (random weight initialization, stochastic gradient descent), they result in relatively consistent prescriptions across different data splits and different runs, similarly to the more deterministic prescriptive methods.

In particular, for the unstructured datasets in Table 11, the standard deviation of Mirrored OCTs is considerably different from the PNNs' in three out of four scenarios. We attribute this to the fact that Mirrored OCTs were likely not able to fully capture the PNNs' complexity and are therefore simpler in their decision-making rules.

### 4.2.3 Interpretability

We discuss now the interpretability of PNNs that is recovered via knowledge distillation of the Mirrored OCTs. In particular, we discuss the unstructured TAVR dataset, for which we can partially recover interpretability, and for the structured diabetes management dataset. Please refer to Appendix A.6 for similar analyses for our other datasets and Appendix A.7 for model visualizations.

**TAVR.** We discuss the multimodal Mirrored OCT from Figure 10 for the TAVR dataset. While the embedding features from the clinical notes are not interpretable, we can still recover some interpretability through

---

[2]For multiple continuous treatments (diabetes dataset), to get the standard deviation for each sample, we first calculate the standard deviation for each drug separately and then we average across the three drugs. For Regress & Compare, since we use XGBoost models, there is no randomness in each split, so the 5 models produce the same prescriptions. This explains why in most of the datasets, Regress & Compare has the lowest standard deviation.

the tabular features selected by the Mirrored OCT. We see that the OCT selects one note feature, as well as the Valve-to-Annular Aortic Valve Area ratio (VDAoVA), age, and difference between the annular area of the patient's native aortic valve and the area of the prosthetic valve being implanted (Area Oversize). We observe that the OCT provides an insight into the prescriptions made by the PNN, even when it is trained on the multimodal data.

**Diabetes management.** We consider and compare the Mirrored OCT from Figure 12 and the Optimal Policy Tree from Figure 13 for diabetes management. Both models were trained with the same data, methods, and parameters as in Section 3.6. For visualization reasons, Figure 12 displays a portion of the tree. From Figure 12, we can see that the features selected by the tree include "HbA1c_mean" (average pre-prescription blood hemoglobin A1C level), "visitNo" (number of visits by the patient), and "pastHbA1c1" (past blood hemoglobin A1C level). For example, if a patient has an average pre-prescription blood hemoglobin A1C level of less than 8.582, has at least 4 visits, and has past hemoglobin A1C levels of less than 9.15, then the patient would be prescribed with treatment "4," which corresponds to 0 units of insulin, 1 unit of metformin, and 0 units of oral blood glucose regulation agents. The Optimal Policy Tree (Figure 13) is simpler, but equally interpretable. We observe that it selects different features; for example, the patient's average and past blood hemoglobin A1C levels are not taken into account.

### 4.3 Real-world deployment potential and challenges

We briefly discuss the potential and challenges associated with the real-world deployment of PNNs and Mirrored OCTs. Given their flexibility and ease of training, both models show strong promise for practical use. That said, implementing prescriptive models—regardless of the specific architecture—poses several challenges, including ethical considerations around automated decision-making. In healthcare settings, for instance, building model trust is a key concern. However, we believe the interpretability of Mirrored OCTs, combined with collaborative model development alongside clinicians, offers a promising path toward deployment. Most importantly, we envision PNNs and Mirrored OCTs as tools to support clinical decision-making, rather than replace it. When the model's recommendation aligns with the clinician's decision, it can offer reassurance; when the two disagree, it can prompt a valuable second look. In a real-world application, the final decision would remain with the clinician, informed by both clinical expertise and the model's guidance.

## 5 Conclusions

With its classification-like feedforward neural network architecture, our PNN framework flexibly handles multimodal data, by easily enabling the incorporation of multiple data sources. Furthermore, it is widely applicable for all treatment scenarios, and has the potential of making a significant impact in a variety of settings, as shown in our extensive real-world experiments. To the best of our knowledge, our method is the first prescriptive model for multimodal data, and it also outperforms or performs comparably to other well-known prescriptive models on unimodal tabular data in all treatment scenarios, without requiring large computational resources.

Our approach is not only shown to perform strongly quantitatively, but also to provide realistic and stable prescriptions. The discrepancy between the prescribed and real-life treatment distributions is comparable to the other prescriptive methods. The small standard deviation of each sample's assignments from the models indicates that the networks are stable and robust to different data splits. Also, PNNs offer the advantage of flexibility since the user can adjust the loss function to provide partial control to the prescriptions, leveraging expert knowledge.

Deep learning methods generally sacrifice interpretability. On unimodal tabular datasets, we are able to recover interpretability through a knowledge distillation approach leveraging interpretable OCT models, and on multimodal datasets, some interpretability may still be recovered. These Mirrored OCTs demonstrate similarly high performance in our real-world experiments, demonstrating that we can maintain high performance without sacrificing interpretability. This recovery of interpretability is critical for real-world deployment of deep learning models.

We conclude that our multimodal deep learning framework, PNNs, offers both flexibility and strong performance, effectively utilizing deep learning to process multimodal data. By integrating multiple data sources, the framework significantly enhances decision-making capabilities. This unified approach demonstrates its potential as a versatile prescriptive tool, well-suited for a wide range of applications.

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

# A   Appendix

## A.1   Embeddings Dimensionality Reduction

We experimented with three variants of text embeddings for training PNNs on the multimodal datasets (TAVR and liver trauma): (1) the full-dimensional embeddings extracted from the pretrained language model, (2) a lower-dimensional representation extracted from an intermediate layer of a fine-tuned classification head, and (3) embeddings reduced using Principal Component Analysis (PCA).

For consistency and fair comparison across methods, all PNNs were evaluated using rewards trained on the full-dimensional embeddings, which preserve the most information and serve as a common reference.

Table 12: Improvement(%) in risk for the experiments with unstructured data (TAVR, liver trauma) when different embeddings are used to train PNNs in multimodal datasets, reported on rewards trained with full-dimensional embeddings.

| Type of embeddings | TAVR | Liver trauma |
|---|---|---|
| Full-dimensional | $26.19 \pm 2.41$ | $19.65 \pm 2.28$ |
| Representation from Classification Head | $\mathbf{42.89 \pm 5.64}$ | $23.94 \pm 3.08$ |
| PCA-reduced | $26.32 \pm 2.57$ | $\mathbf{25.25 \pm 3.00}$ |

Across both datasets, dimensionality reduction leads to improved performance compared to using full-dimensional embeddings. For TAVR, representations extracted from the classification head yield the best results, suggesting that supervised fine-tuning can produce more informative and stable embeddings. In the liver trauma dataset, both classification head and PCA-reduced embeddings outperform the full-dimensional version, with PCA achieving the highest improvement. These results suggest that dimensionality reduction can improve model stability and performance, especially when the original embedding dimension is high relative to dataset size.

## A.2   PNN Hyperparameters

We specify and finetune the following PNN hyperparameters:

- **Number of layers of the network.** We experiment with both shallow and deep networks. Though conclusions differ based on the dataset, in general we observe that deeper networks do not necessarily improve results.

- **Number of nodes at each layer.** The size of the dataset closely affects this hyperparameter. Typically more nodes per layer are used with larger datasets that also include more features.

- **Batch size.** This parameter determines the number of samples used in each forward pass of the network and for backpropagation, where the network parameters are updated after each batch passes through the network. It also affects the training speed, since too many batches can slow down the training process. Again, there is a correlation between batch size and the size of our data; larger batch sizes are employed for larger datasets.

- **Learning rate.** The learning rate is an important parameter of the training process, since it defines how steep the descent is at each step of the gradient descent algorithm during training. After experimentation for each dataset, we find an appropriate learning rate that is not too big so that the algorithm becomes stuck in local optima but also not too small so that convergence is too slow.

- **Weight decay.** This hyperparameter scales an $L_2$-regularization term of the network weights that is added to the objective function to prevent them from taking too large values. Since our data is normalized, we observe that lowering the weight decay coefficient and relaxing the weights are actually beneficial and do not result in overfitting.

- **Number of epochs.** The number of epochs is a particularly hard parameter to tune, since we want to prevent overfitting but also allow the training to continue until sufficient convergence. For this reason, we employ early stopping, a technique that is adaptive to each specific training process and terminates training when a fluctuation in the validation loss is observed. Such a fluctuation indicates that the network is no longer improving in out-of-sample data generalization.

### A.3 Additional tables

Table 13: Treatment scenarios, data splits, and counterfactual estimation methods used for each structured dataset.

| Dataset | Treatment scenario | Counterfactual estimator |
|---|---|---|
| Groceries pricing | Single continuous | XGBoost Classifier (direct method) |
| Splenic injuries treatment | Multiple discrete | Random Forest Classifier (doubly-robust) |
| REBOA in blunt trauma patients | Binary | Random Forest Classifier (doubly-robust) |
| Diabetes management | Multiple continuous | Random Forest Regressor (doubly-robust) |

## A.4 Positivity plots

Figure 3: TAVR propensity score distributions.

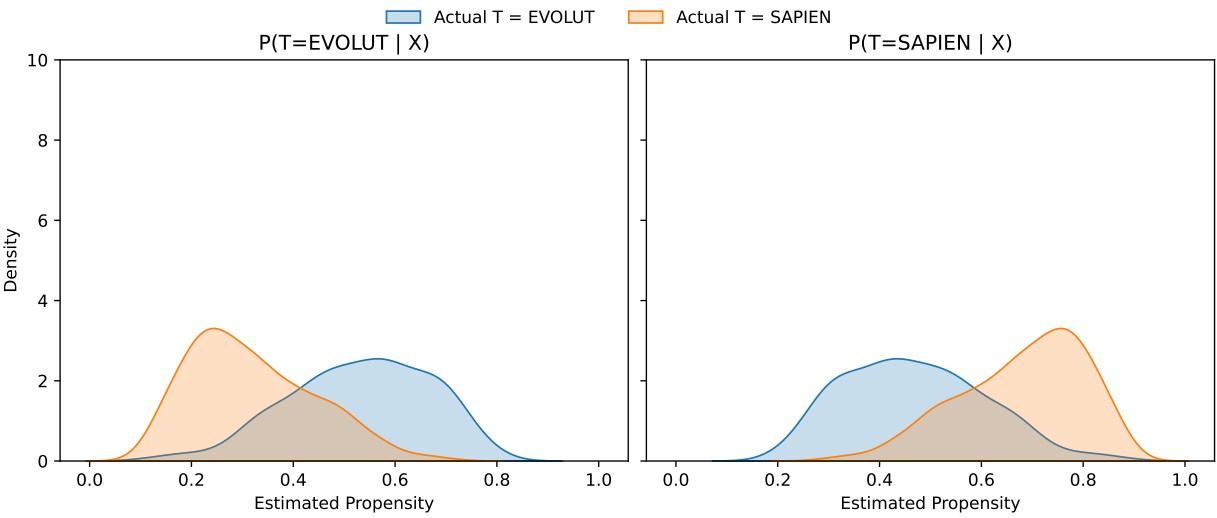

Figure 4: Liver injury propensity score distributions.

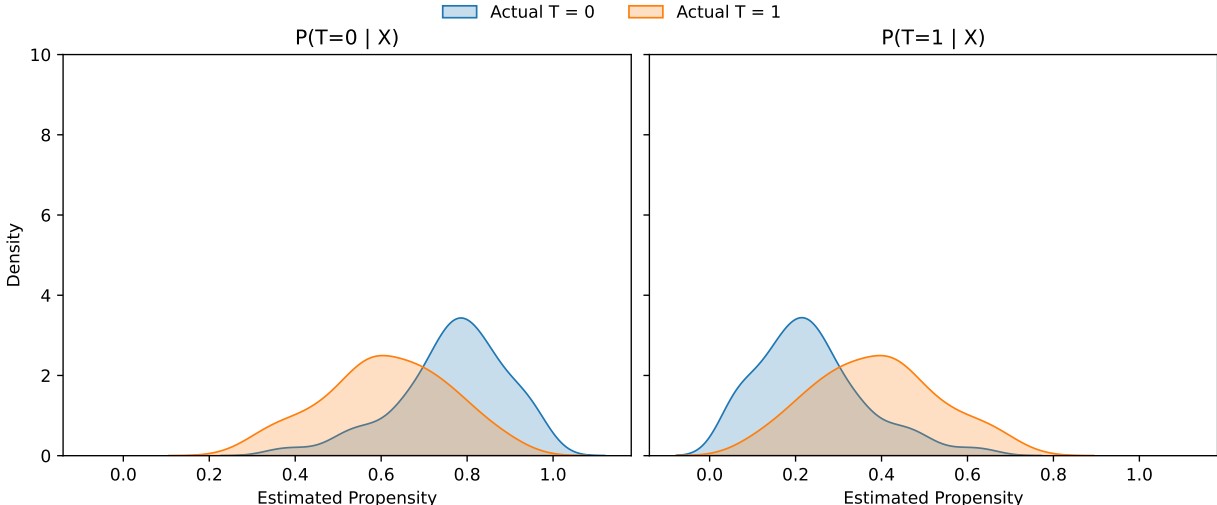

Figure 5: Diabetes propensity score distributions.

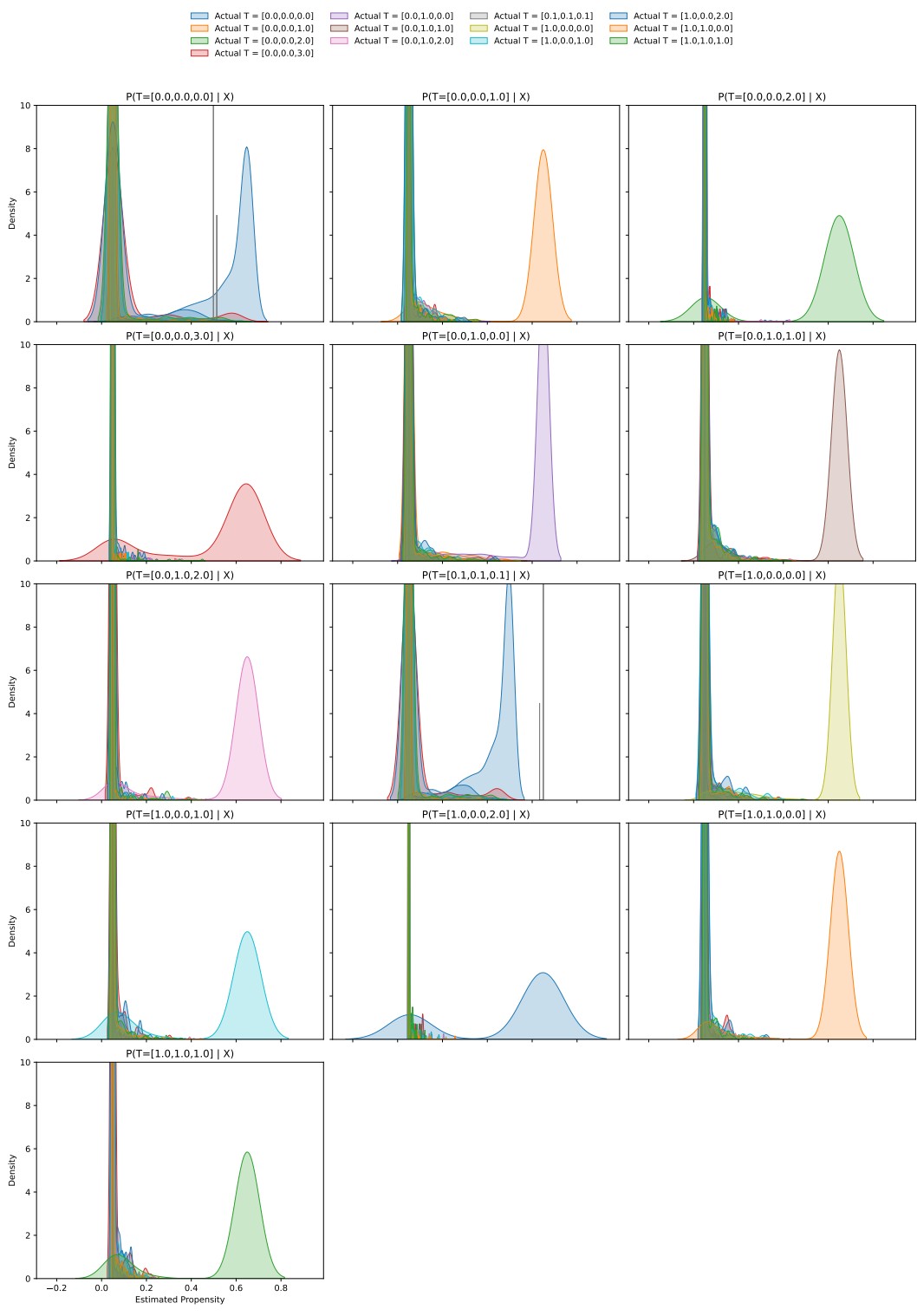

Figure 6: Groceries propensity score distributions.

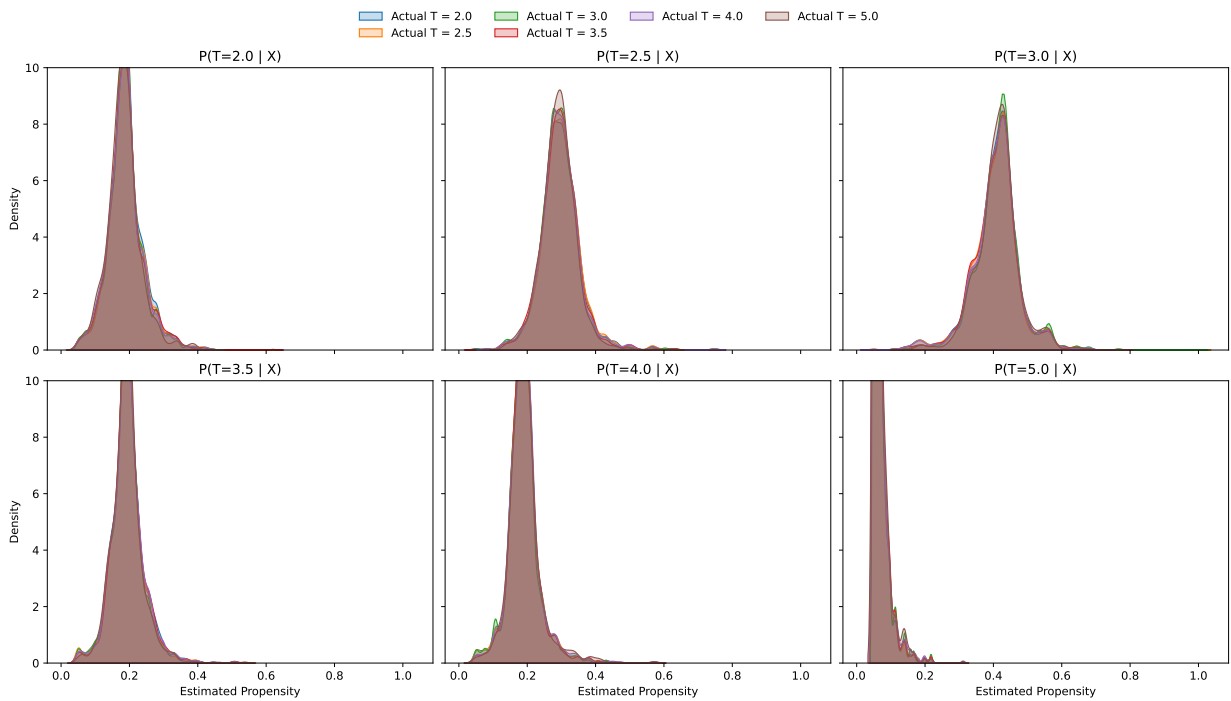

Figure 7: Spleen propensity score distributions.

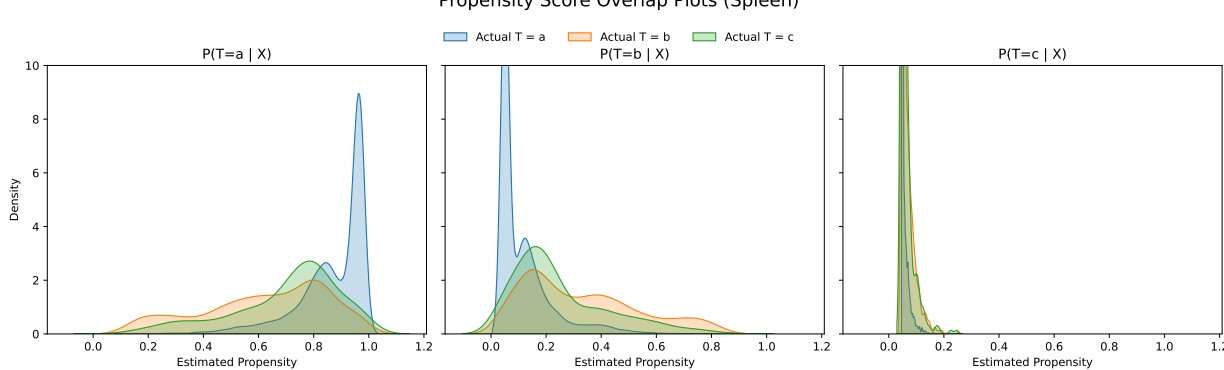

Figure 8: REBOA propensity score distributions.

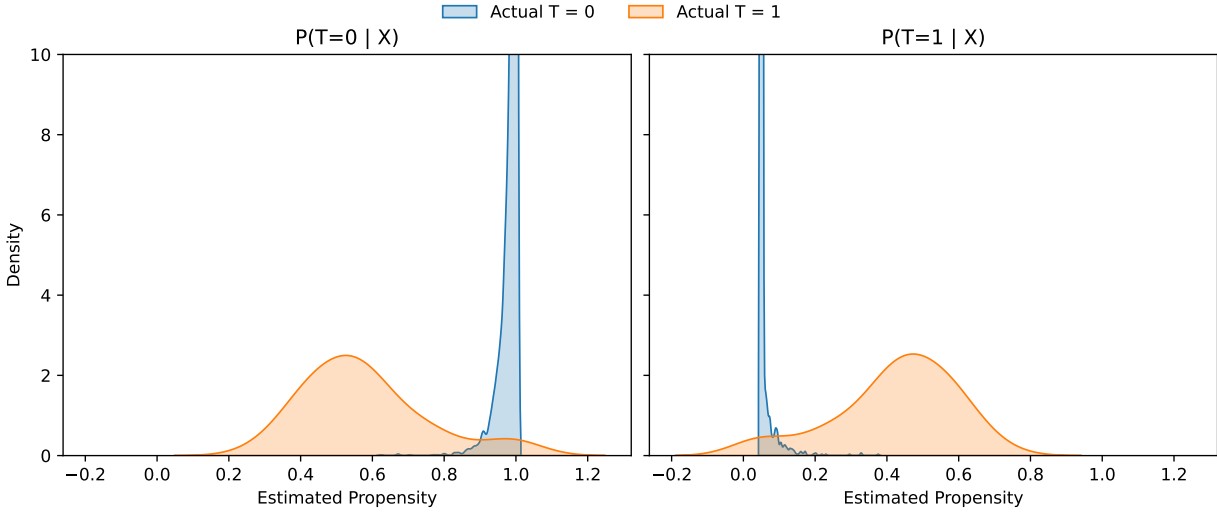

## A.5 Treatment distribution

Figure 9: Treatment distribution grid.

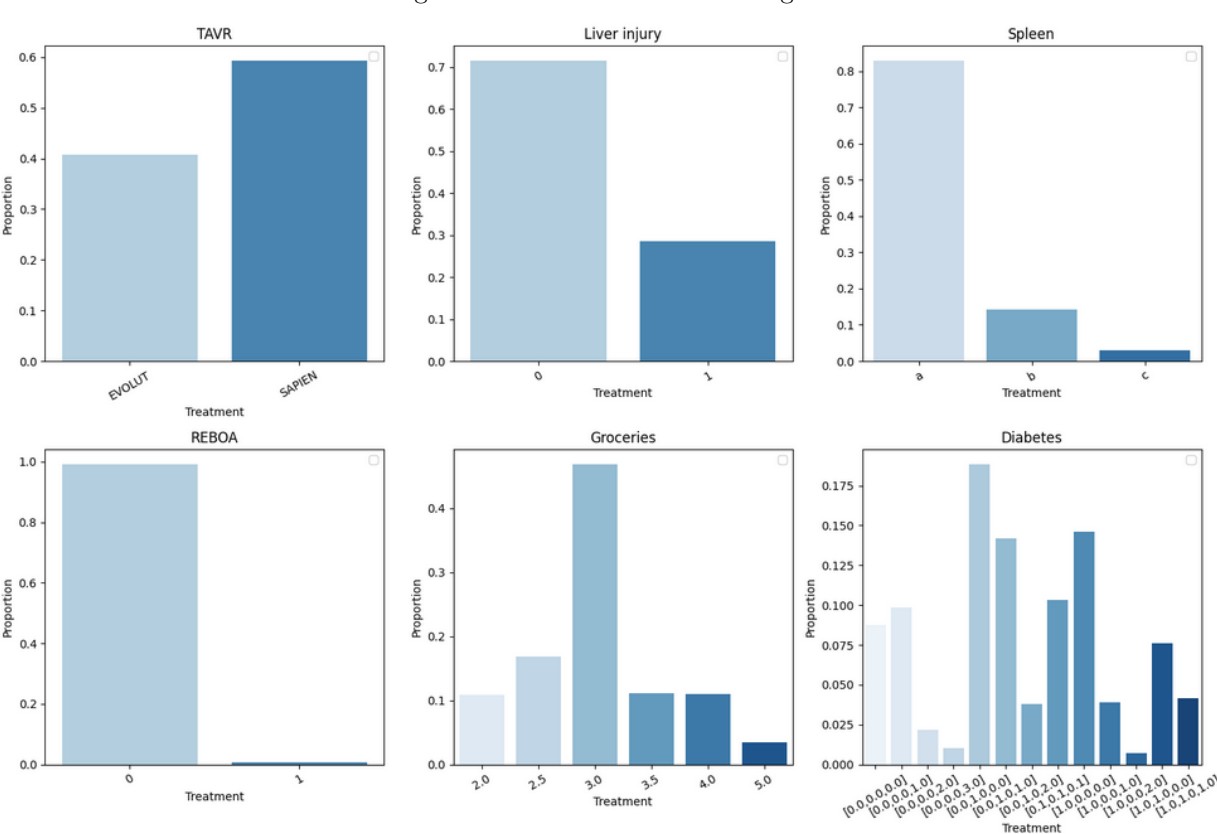

The treatments for the different datasets are the following:

- **TAVR**: There are two potential valves, SAPIEN and EVOLUT.

- **Liver injury**: 0 corresponds to no surgery, and 1 corresponds to prescribing surgery.

- **Diabetes**: Each vector contains the dosage of insulin, metformin and oral treatment in that order.

- **Groceries**: Each value corresponds to the prescribed price, in dollars.

- **Spleen**: Treatment "a" corresponds to observation, treatment "b" corresponds to splenectomy and treatment "c" corresponds to angioembolization.

- **REBOA**: 0 corresponds to no procedure, and 1 corresponds to prescribing the procedure.

### A.6 Extension of interpretability of unstructured and structured real-world datasets

**Liver trauma.** We discuss the multimodal Mirrored OCT from Figure 11 for the liver injury dataset. The embedding features from the clinical notes are not nearly as interpretable, but we can still interpret the tabular features selected by the Mirrored OCT. We see that the OCT selects two note features, as well as whether the patient had other symptoms with the circulatory or respiratory system and whether the patient had pain in their throat or chest.

**Groceries.** We compare the resulting Mirrored OCT from Figure 14 and the Optimal Policy Tree (OPT) from Figure 15. The features selected by both the Mirrored OCT and the Optimal Policy Tree are similar, with the most prominent ones being the homeowner status, age, income range, household status, and marital status. An interesting difference between the two trees is the number of distinct prescriptions selected: the OCT only selects 3 of the 6 pricing options – prediction classes 0, 1, and 5 (which correspond to prices USD $2.00, $2.50, and $5.00) – whereas the OPT prescribes more of the possible options. The OCT's strategy seems to therefore select low prices for lower-income households, compared to high prices for households with more financial stability.

**Splenic injuries treatment.** We compare the Mirrored OCT displayed in Figure 16 with an Optimal Policy Tree (OPT) in Figure 17. There are three possible treatments: simple observation (treatment 0 of the OCT, "a" for the OPT), splenectomy (treatment 1 of the OCT, "b" for the OPT) and angioembolization (treatment 2 of the OCT, "c" for the OPT). The OCT prescribes the first and third options while the OPT prescribes all three. The two trees split on similar features, although at different levels of the tree; these features include "totalgcs" (Glasgow coma scale), the state (grade) of the spleen (injury severity on a scale of increasing acuteness from 1 to 5), diabetes, and smoking status. However, the Mirrored OCT splits on SBP (systolic blood pressure), TBI (Traumatic brain injury) and intubation status, while the OPT splits on BMI, respiratory rate, and history of cirrhosis.

**REBOA in blunt trauma patents.** We compare the Mirrored OCT from Figure 18 and the Optimal Policy Tree (OPT) from Figure 19.

From Figure 18, we clearly see that the important features include, among others, "pneumothorax_1" (whether the patient has pneumothorax, a collection of air outside the lung but within the pleural cavity), "gcs" (Glasgow coma scale), SBP (systolic blood pressure), and pulse rate. We observe that the OPT (Figure 19) only splits on SBP and pulse rate, indicating that the resulting Mirrored OCT is considering more factors to make the final prescription, which is potentially closer to real life scenarios.

### A.7 Mirrored OCTs on all real-world datasets

Further examples of Mirrored OCTs of maximum depth 7 for the TAVR, liver trauma, diabetes, groceries, splenic injuries, and REBOA datasets can be found under `https://drive.google.com/drive/folders/12XNOQllyzVQEFFguHvkcAQ-7Psp9KriQ`.

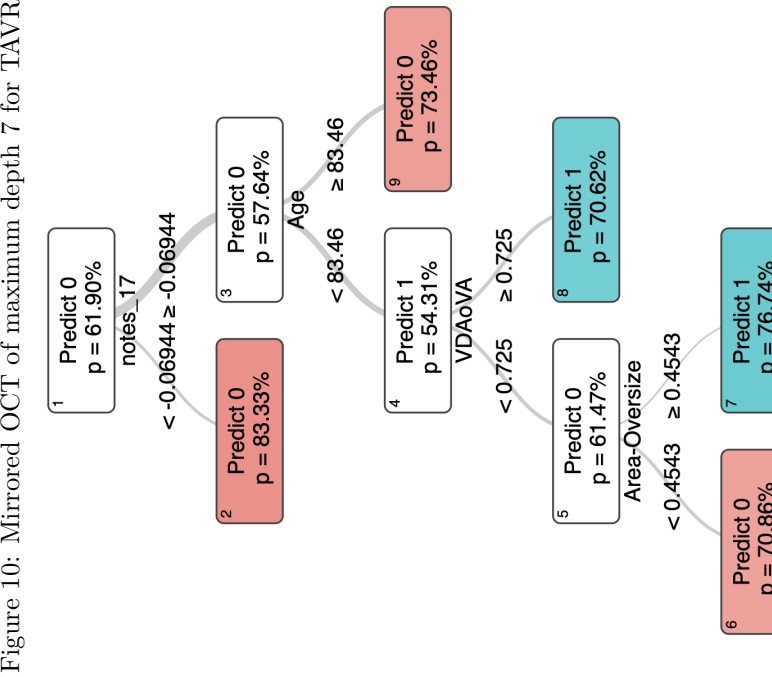

Figure 10: Mirrored OCT of maximum depth 7 for TAVR.

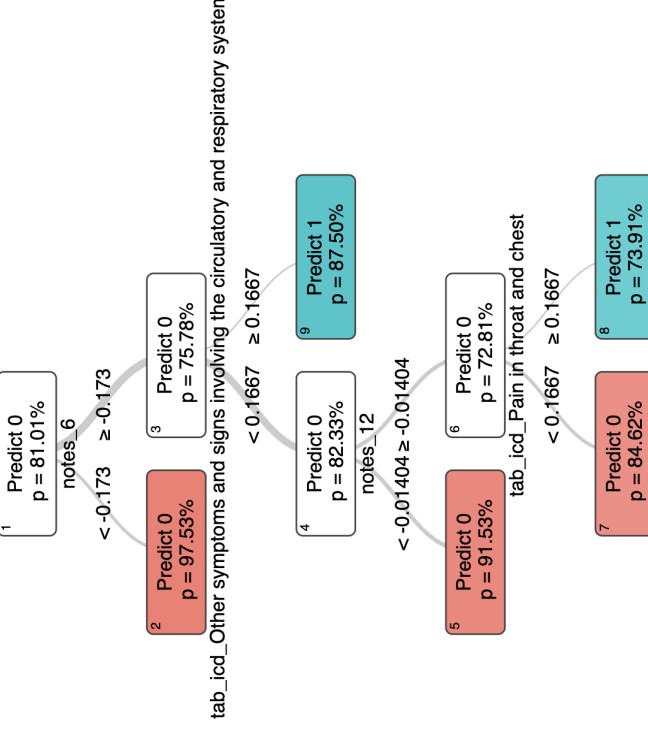

Figure 11: Mirrored OCT of maximum depth 7 for liver trauma.

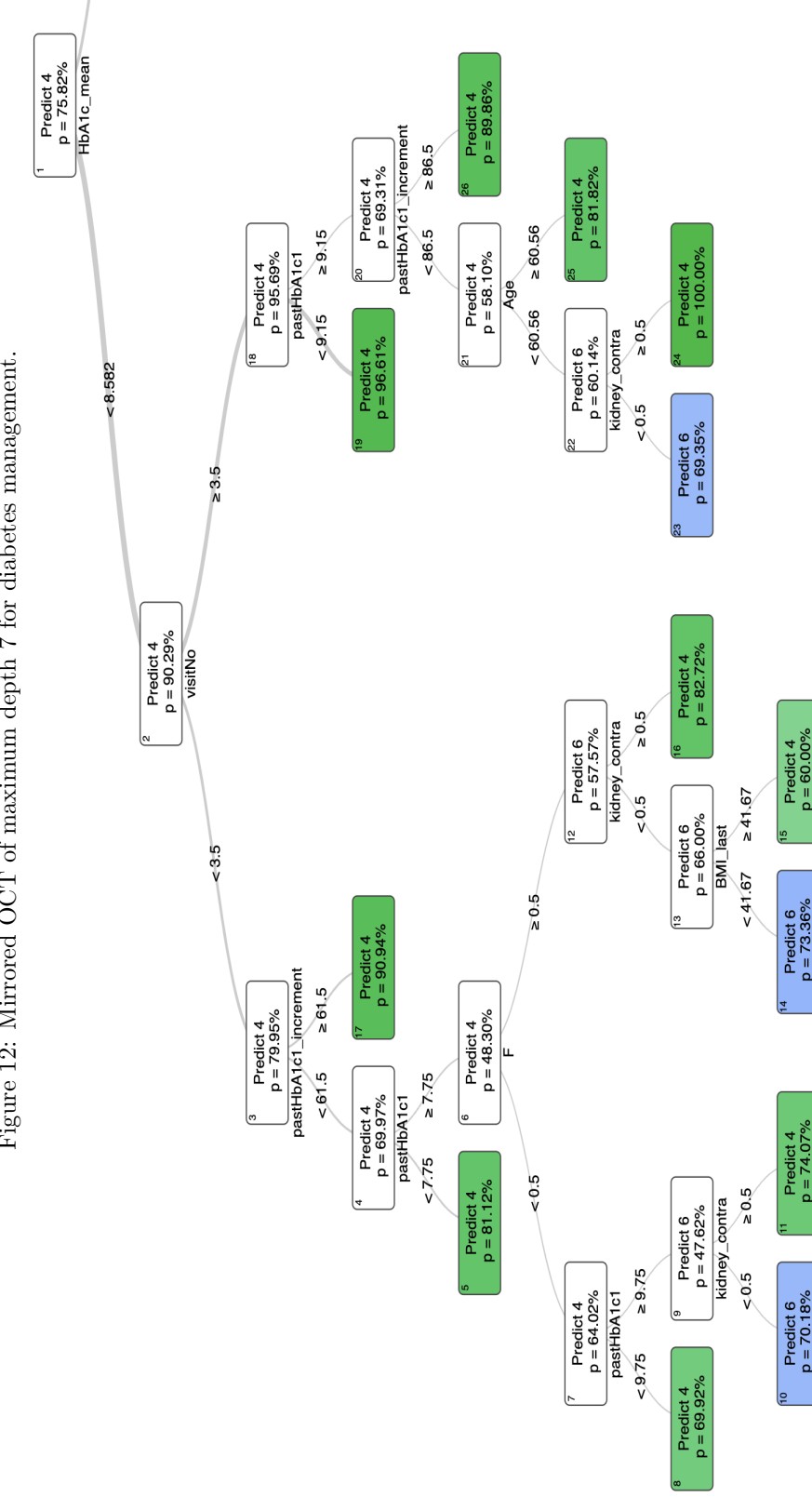

Figure 12: Mirrored OCT of maximum depth 7 for diabetes management.

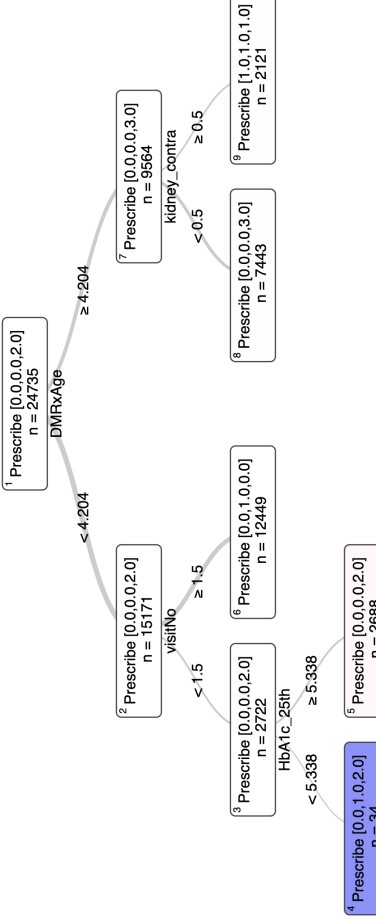

Figure 13: Optimal Policy Tree of maximum depth 7 for diabetes management.

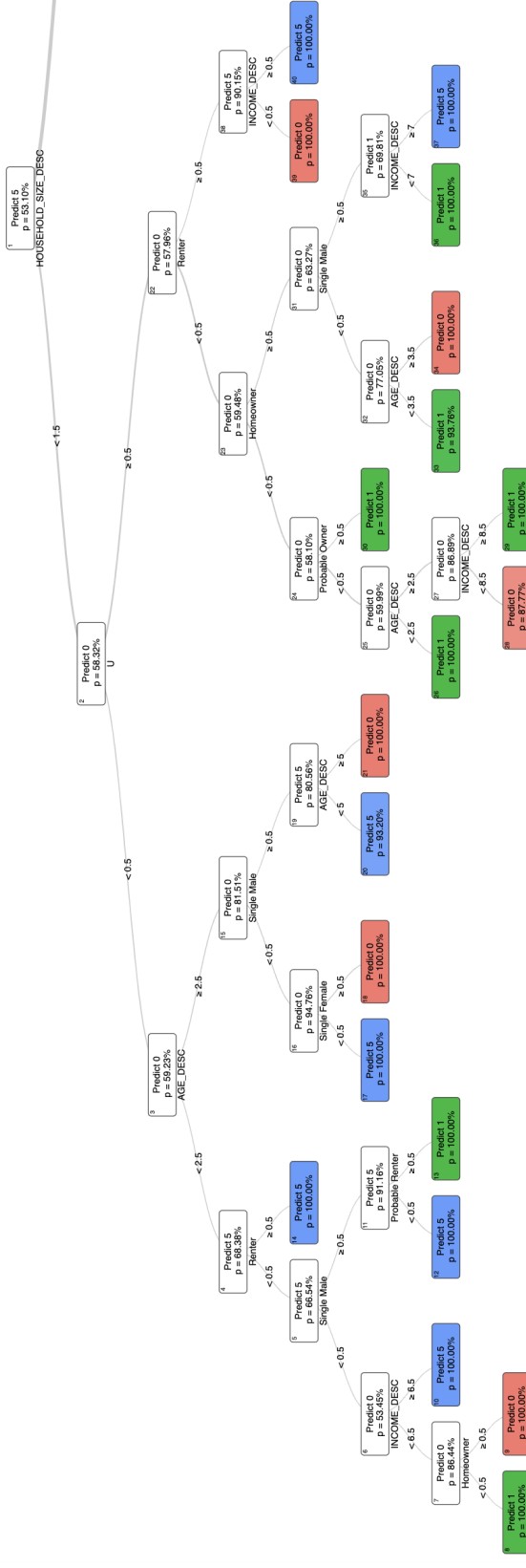

Figure 14: Mirrored OCT of maximum depth 7 for groceries.

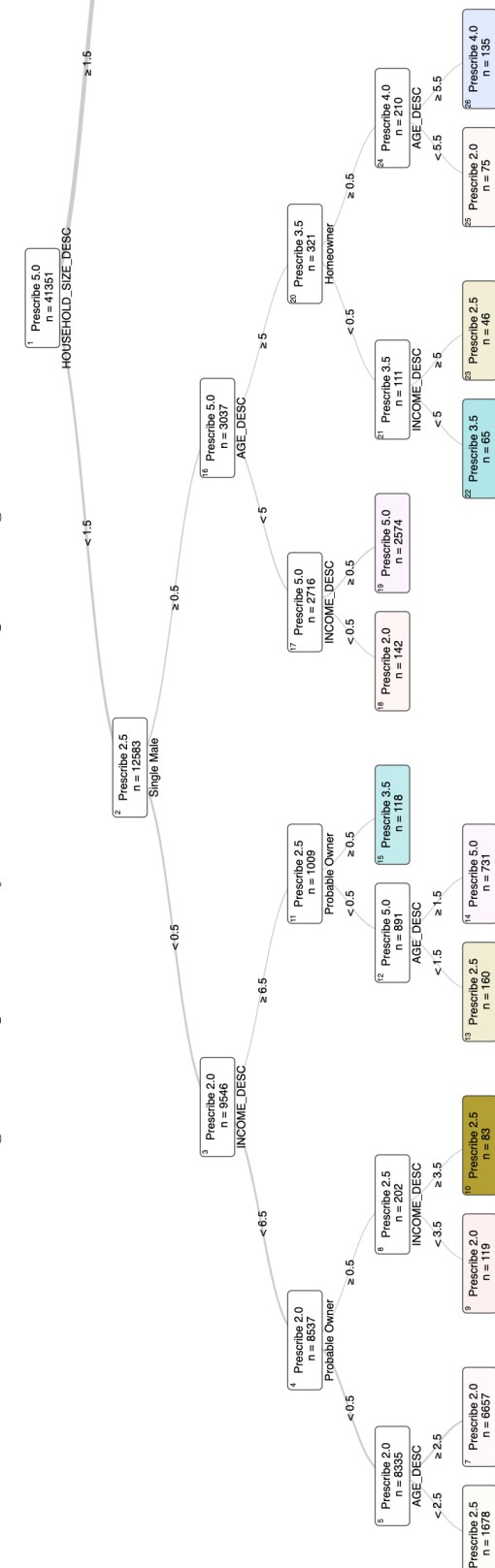

Figure 15: Optimal Policy Tree of maximum depth 7 for groceries.

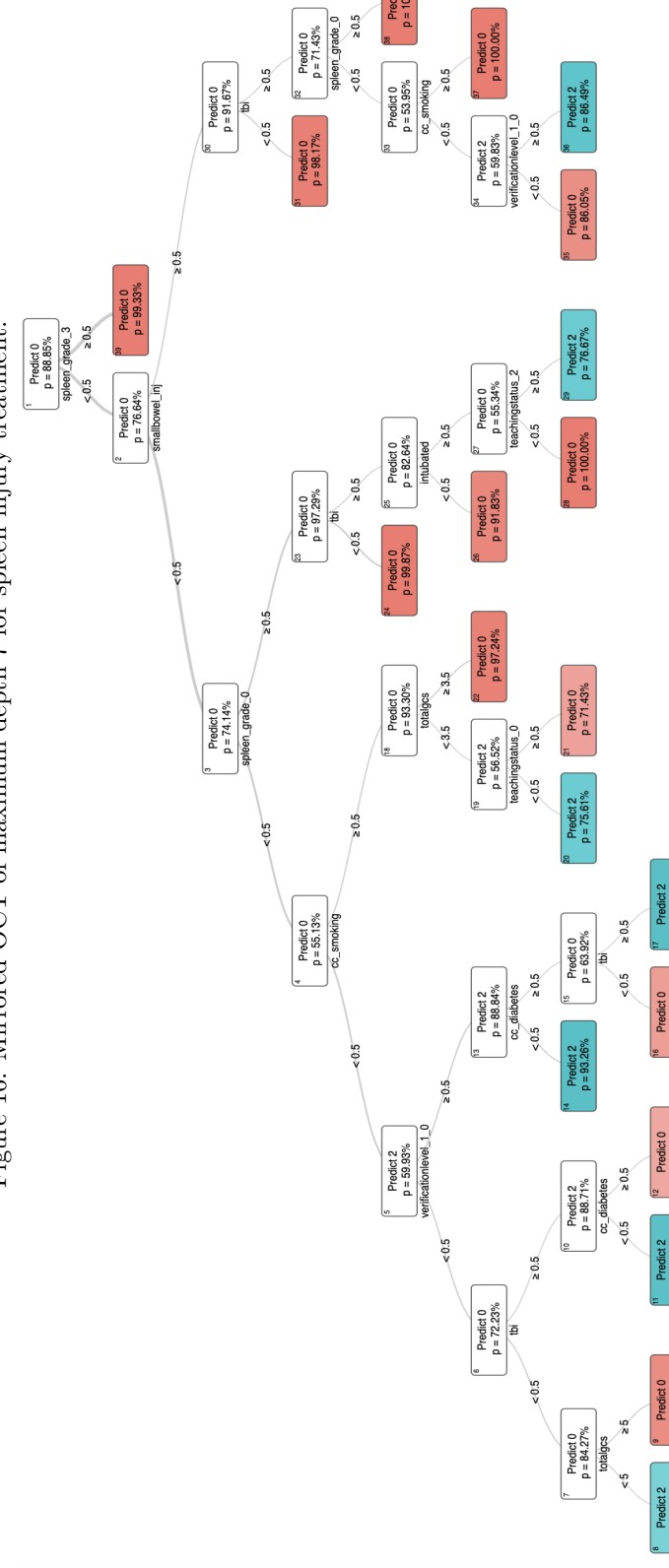

Figure 16: Mirrored OCT of maximum depth 7 for spleen injury treatment.

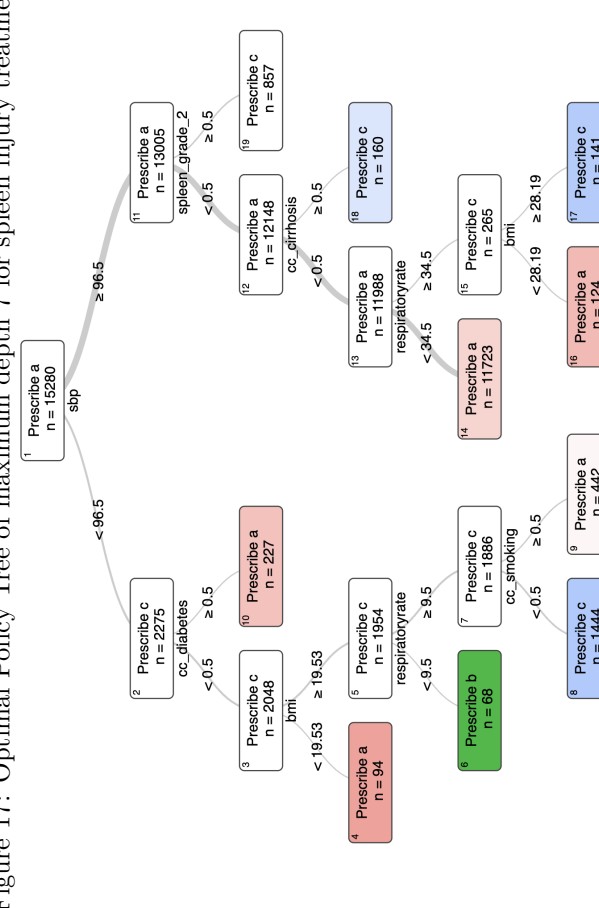

Figure 17: Optimal Policy Tree of maximum depth 7 for spleen injury treatment.

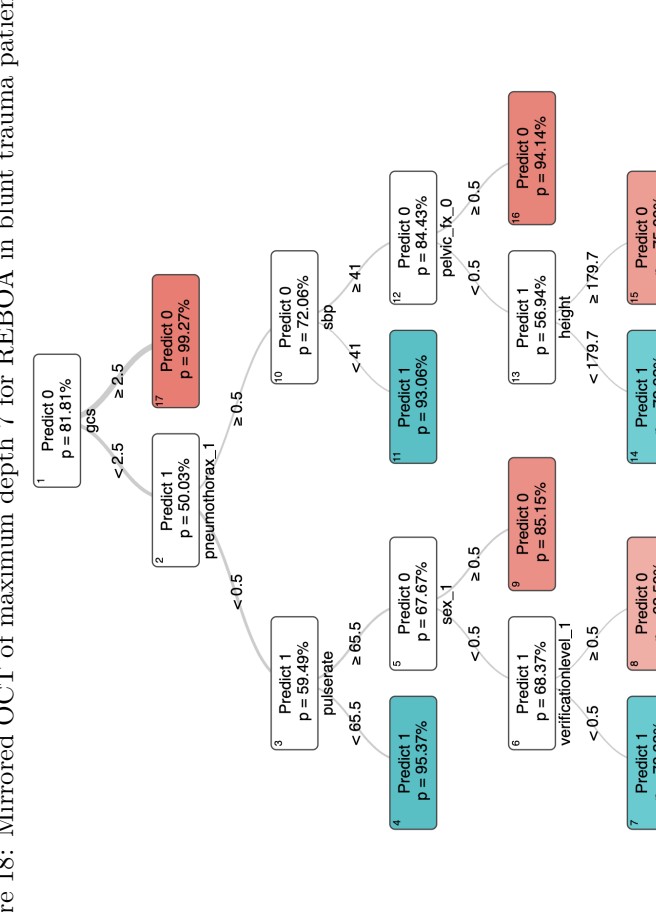

Figure 18: Mirrored OCT of maximum depth 7 for REBOA in blunt trauma patients.

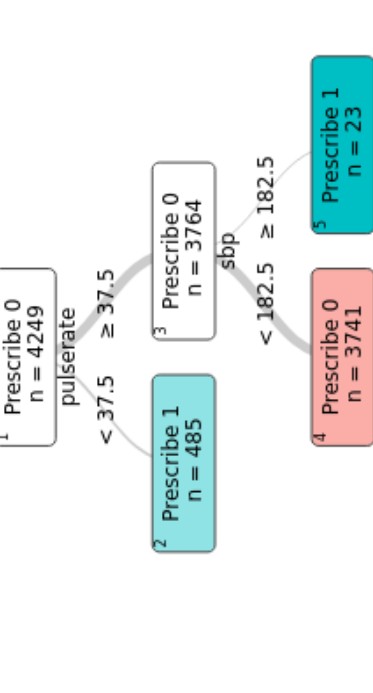

Figure 19: Optimal Policy Tree of maximum depth 7 for REBOA in blunt trauma patients.

