# OpenReview forum: "Multimodal Prescriptive Deep Learning"
_TMLR — Rejected by TMLR_

### Review · Reviewer_1nqP · 2025-02-24

**Summary Of Contributions:**

Overall, the paper presents a novel framework for prescribing treatments using multimodal data via what the authors call Prescriptive Neural Networks (PNNs). While the idea of combining deep learning with counterfactual estimation is compelling, several technical and mathematical issues arise upon closer inspection.

**Audience:**

Yes

**Broader Impact Concerns:**

NAN

**Claims And Evidence:**

Yes

**Requested Changes:**

See weakness.

**Strengths And Weaknesses:**

Strengths:
The framework is designed to handle various treatment types. Mathematically, the core objective is to find the optimal treatment by solving:
  t^* = \arg\min_{t \in T} \; \Gamma(x,t),
  where Gamma(x,t) represents the estimated outcome under treatment t. This flexibility is a significant advantage.

The submission employs a doubly robust estimator for counterfactual outcomes, defined as:
  \Gamma_{i,t} = \hat{y}_{i,t} + \mathbf{1}\{t_i=t\}\frac{1}{p_{i,t}}(y_i - \hat{y}_{i,t}),
  which helps mitigate treatment assignment bias—a common challenge in observational studies.

Recovering interpretability by fitting Optimal Classification Trees (OCT) to mimic the PNN's prescriptions bridges the gap between deep learning’s black-box nature and the need for transparent decision-making. This approach provides a practical compromise between accuracy and interpretability.

Weaknesses

Sensitivity to Propensity Scores:   In the doubly robust estimator, if the propensity scores p_{i,t} are very small, the term \frac{1}{p_{i,t}} may induce high variance. The authors should discuss potential safeguards, such as clipping p_{i,t}, to stabilize the estimates.

 While the softmax function is used to approximate the non-differentiable indicator function, the impact of the smoothing parameter is not explored. Introducing a temperature parameter \tau in the softmax:
  \sigma_t(z; \tau) = \frac{e^{z_t/\tau}}{\sum_{j \in T} e^{z_j/\tau}},
  and analyzing its effect on convergence and performance could provide deeper insights.

The paper handles continuous treatments by discretizing the treatment space. However, there is no rigorous analysis of the error introduced by this process, such as the approximation error:
Error = | y(x_i, t) - y(x_i, \tilde{t}) |,
  where \( \tilde{t} \) is the discretized treatment. A formal discussion on how the discretization impacts overall performance is needed.

The decision to use PCA for dimensionality reduction on high-dimensional embeddings may not be optimal, as it is linear in nature. Nonlinear methods (e.g., autoencoders) might capture the inherent complexity of unstructured data better, which the authors could consider.

Although the empirical results are promising, the paper lacks a detailed discussion on the scalability of the proposed method, particularly in high-dimensional and multimodal settings. An analysis of computational complexity would benefit readers assessing its practical deployment.

While the OCT approach provides interpretability, there is no formal quantification of the trade-off between interpretability and performance. A more detailed analysis of the approximation error between the PNN and OCT predictions would be valuable.

---

### Review · Reviewer_7LhU · 2025-03-14

**Summary Of Contributions:**

This paper introduces Prescriptive Neural Networks (PNNs), a novel multimodal prescriptive deep learning framework that integrates machine learning and optimization to directly learn prescriptions from data. This is a method capable of handling multimodal data, extending prescriptive analytics beyond structured tabular settings.

### Key Contributions

1. **Multimodal Prescriptive Learning**
   - Proposes PNNs as a deep learning-based approach to prescriptive decision-making, capable of leveraging both structured and unstructured data.
   - Demonstrates that multimodal PNNs significantly outperform unimodal models, particularly in clinical decision-making tasks.

2. **Empirical Validation Across Multiple Treatment Scenarios**
   - Evaluates PNNs on six real-world datasets, covering all major treatment assignment types:
     - Binary treatments (e.g., REBOA for trauma patients).
     - Multiple discrete treatments (e.g., splenic injury management).
     - Single and multiple continuous treatments (e.g., diabetes drug prescriptions and groceries pricing).
   - On multimodal medical datasets, PNNs reduce estimated:
     - Postoperative complications in TAVR by 32%.
     - Mortality rates in liver trauma injuries by over 40%.

3. **Comparison to Existing Prescriptive Models**
   - Benchmarks against state-of-the-art prescriptive methods, including:
     - Regress & Compare,
     - Causal Forests,
     - Optimal Policy Trees.
   - Demonstrates that PNNs outperform or match these baselines across structured datasets.
   - However, key causal baselines (e.g., meta-learners, deep causal models) are missing, which could further validate effectiveness.

4. **Interpretability Through Knowledge Distillation**
   - Proposes Mirrored Optimal Classification Trees (Mirrored OCTs) to approximate PNN prescriptions, improving interpretability for structured datasets.
   - Finds that Mirrored OCTs maintain performance with only a 1.38% decrease in outcome improvement.
   - However, interpretability remains limited in multimodal cases, where text/image embeddings are used.

5. **Robustness and Stability**
   - PNNs exhibit consistent performance across randomized data splits, suggesting stable prescriptions.
   - However, the methodology does not explicitly handle confounding bias or evaluate causal validity.
   - The lack of a formal causal model (e.g., DAGs, positivity checks, counterfactual validity testing) is a potential limitation.

**Audience:**

Yes

**Broader Impact Concerns:**

-

**Claims And Evidence:**

No

**Requested Changes:**

## Suggested improvements
* Incorporate causal inference theory:
   - Explicitly modeling confounding using graphical models, or
   - Discussing positivity, ignorability, and treatment overlap within a formal causal inference framework.
* Reformat evaluation metrics to use confidence intervals and clearly indicate statistically significant differences.
* Consider cross-validation as a more robust evaluation method.
* Streamline presentation by:
   - Removing redundant sections (e.g., detailed hyperparameter tuning, MLP explanation).
   - Moving standard details (e.g., training accuracy) to an appendix.
* Include an ablation study on PCA’s impact on performance and stability.

**Strengths And Weaknesses:**

## Strengths

- Introduces Prescriptive Neural Networks (PNNs), the first multimodal prescriptive model integrating machine learning and optimization.
- Expands prescriptive analytics to multimodal settings, demonstrating strong practical relevance in medical decision-making.
- Outperforms or matches state-of-the-art prescriptive models on four structured datasets.
- Applicable to all treatment scenarios: binary, discrete, single continuous, and multiple continuous treatments.

## Weaknesses

### Lack of causal inference foundations
My main concern is that the paper does not reference foundational causal inference theory. There is no graphical model, no explicit assumptions about confounders, and no discussion on treatment overlap (e.g., is positivity guaranteed?). The paper would benefit significantly from integrating insights from Rubin’s Potential Outcomes Framework or Pearl’s Structural Causal Models, which provide well-established tools for handling confounding bias and ensuring validity of counterfactual estimates.

### Presentation and organization
Certain sections include excessive details that may not add value for an ML journal audience. Examples include:
- Section 2.4, which provides a tutorial-style explanation of multinomial logistic regression—this could be shortened or omitted.
- Figure 1, which illustrates a generic MLP architecture, does not add significant insight and could be removed. Moreover, Quiza and Davim (2011) should not be the canonical reference for MLPs.
- Section 3.2 (Hyperparameters Tuning), which describes standard neural network hyperparameters in detail, would be more appropriate in an appendix.

### Methodological concerns
- **Evaluation metrics:**
  - Tables 1 and 4 present results using standard deviation instead of standard errors or confidence intervals (CIs). Since overlapping confidence intervals suggest non-significant differences, it would be better to report standard errors and explicitly indicate statistically significant improvements.
  - The number of experiments appears to be \( N = 5 \)—could the authors clarify whether this is sufficient for robust statistical comparison?
  - If the performance differences overlap significantly, it would be helpful to highlight methods that clearly outperform others.
- **Missing baselines:**
  - DragonNet, TARNet, T-learners, X-learners, S-learners (Kunzel et al. (2019), "Metalearners for Estimating Heterogeneous Treatment Effects."). Please either discuss why they might be relevant or compare against them.
- **Data splitting strategy:**
  - The paper justifies a 50-50 train-test split in the name of fairness, but the definition of fairness in this context is unclear.
  - A cross-validation approach would likely provide more robust and less biased performance estimates. Could the authors comment on why it was not used?

- **Dimensionality reduction (PCA):**
  - The authors state that PCA is optionally applied to improve stability and performance. Could this claim be validated with an ablation study to assess whether PCA meaningfully improves results?

- **Table 3 (Training Accuracy):**
  - The purpose of reporting training accuracy is unclear. If its inclusion is necessary, it should be moved to the appendix—or, if not particularly insightful, removed altogether.

## Additional observations and questions

- 1.1 Regress & Compare: This section is missing an appropriate reference.
- 2.2 “Numerical features are normalized”—Could the authors clarify how the normalization is performed?
- Table formatting: Some tables contain a lot of content but could be streamlined to focus on key results.

---

### Review · Reviewer_NX5V · 2025-03-17

**Summary Of Contributions:**

The paper introduces Prescriptive Neural Networks (PNNs), a deep learning framework designed to prescribe optimal treatments by leveraging multimodal data. PNNs integrate optimization techniques with machine learning to generate prescriptions aimed at improving patient outcomes. The authors evaluate PNNs on two multimodal medical datasets (TAVR and liver trauma) and four unimodal datasets (diabetes management, grocery pricing, splenic injuries, and REBOA). The results indicate that PNNs significantly enhance estimated outcomes compared to existing prescriptive models. Additionally, the study introduces Mirrored Optimal Classification Trees (Mirrored OCTs) to enhance interpretability by distilling PNN prescriptions into transparent decision trees, maintaining comparable performance

**Audience:**

Yes

**Claims And Evidence:**

Yes

**Requested Changes:**

A detailed ablation study examining the impact of multimodal inputs, counterfactual estimation choices, and architectural modifications would be beneficial.
The paper should discuss potential biases in real-world prescription data, such as demographic disparities.
Visualizing the prescribed treatments and their expected impact would improve interpretability for clinical practitioners.


Here are some additional questions the authors can address to strengthen their paper:

Methodological Questions
- Counterfactual Estimation Assumptions: How does treatment assignment bias impact the accuracy of counterfactual estimates, even with doubly robust estimation? Have you tested alternative counterfactual estimation methods (e.g., GAN-based counterfactuals or causal deep learning models)?
- Model Performance and Generalization: How does PNN performance vary with dataset size, particularly for smaller datasets where deep learning models may struggle with overfitting? Have you evaluated PNNs on external datasets beyond those presented to assess generalizability? How do PNNs perform under distribution shifts or adversarial conditions (e.g., missing data, biased treatment distributions)?
- What are the computational requirements of training and deploying PNNs on large-scale healthcare datasets? How do training times and resource consumption compare with tree-based methods, especially for real-time applications?

Theoretical and Interpretability Questions
- Theoretical Guarantees: Can you provide formal proofs or theoretical bounds on the optimality of PNN-prescribed treatments? Under what conditions does the softmax-based treatment assignment converge to an optimal policy?
- Interpretability and Trustworthiness: What strategies can be employed to make PNNs more interpretable beyond Mirrored OCTs? How do domain experts (e.g., clinicians) perceive the interpretability of Mirrored OCTs, and do they trust their recommendations?

Real-World Deployment and Ethical Considerations
- Clinical and Practical Implementation: What challenges exist in integrating PNNs into clinical decision-making workflows? How would the model handle treatment constraints (e.g., unavailable drugs, insurance limitations, ethical concerns)? How do PNN prescriptions compare to human expert decisions in clinical practice?

**Strengths And Weaknesses:**

Some of the key strengths and weakness of the paper are as follows:

The paper presents its claims clearly:
- PNNs are the first multimodal prescriptive method.
- They outperform or match state-of-the-art prescriptive models.
- Interpretability is partially recovered using Mirrored OCTs.
The experimental results support these claims with rigorous comparisons and statistical evaluations. However, while the performance improvements are well-documented, the paper does not provide sufficient discussion on real-world clinical adoption challenges, such as regulatory considerations or integration into healthcare workflows.

The methodology is well-structured and covers key components:
- Data processing: Embedding extraction for structured and unstructured data.
- Counterfactual estimation: Direct and doubly robust approaches to estimate treatment effects.
- Prescription learning: A softmax-based feedforward neural network optimizes treatment assignments.
- Interpretability recovery: Knowledge distillation via Mirrored OCTs.
While these metrics are appropriate, the study lacks an external validation dataset to assess generalizability in new settings. Additionally, computational efficiency is not discussed, which is relevant for large-scale deployment.

The theoretical foundations of PNNs are sound, leveraging deep learning for prescriptive decision-making. The use of softmax for treatment selection aligns with modern classification-based approaches.
However, the paper does not provide formal proofs regarding the convergence or optimality of PNN prescriptions. Assumptions underlying counterfactual estimation methods (especially treatment assignment bias) are not critically examined. Providing a deeper theoretical analysis would strengthen the framework’s credibility.
The experimental setup is robust, featuring comparison to baseline models, validation across multiple train-test splits to ensure stability, and evaluation on datasets covering diverse treatment types (binary, discrete, and continuous treatments).

To summarize we can note these as
**Strengths**:
- Comprehensive comparisons demonstrating PNNs’ effectiveness.
- Consistent performance gains in multimodal datasets.
- Interpretability analysis through Mirrored OCTs.

**Weaknesses**:
- No real-world implementation study assessing deployment feasibility.
- Computational cost analysis is missing, which is essential for assessing scalability.

---

### Author Response · Authors · 2025-02-24

Dear Reviewer 1nqP,

Thank you for the timely feedback. We think that you may have inputted a different paper's review into the Strengths and Weaknesses section. Would you be able to submit the one for our PNN paper? Thanks!

---

### Author Response · Authors · 2025-03-27
**Thank you for the feedback, request for short extension of one week**

Hello,

We first want to thank the reviewers for their detailed comments and thorough feedback. We are currently working on implementing the proposed changes and improving the submitted paper. We were expecting to have finished with our edits by the deadline (this coming Sunday 03/30, 2 weeks after we received the 3rd review), which is why we had not already reached out for an extension, but to ensure that we complete all the suggested edits as carefully as we want, we unfortunately think that we need more time. We are kindly asking for a short extension of one week (Sunday 04/06) before we submit our full response.

Please let us know if this is feasible. Thank you in advance!

---

### Author Response · Authors · 2025-03-30
**Response to feedback from Reviewer 1nqP**

# Response to Reviewer 1nqP

Thank you very much for your thorough feedback and thoughtful comments. We have included responses to all of your points, and have also noted the areas that we are currently still working on. We are expecting to have finalized the edits as soon as possible, this coming week. The completed edits are in the current revision.

Sensitivity to Propensity Scores: In the doubly robust estimator, if the propensity scores $p_{i,t}$ are very small, the term $\frac{1}{p_{i,t}}$ may induce high variance. The authors should discuss potential safeguards, such as clipping $p_{i,t}$, to stabilize the estimates.

**Response: Thank you for the feedback. We are currently working on clipping the propensity estimates that have values below 0.05 and will follow up once completed.**

While the softmax function is used to approximate the non-differentiable indicator function, the impact of the smoothing parameter is not explored. Introducing a temperature parameter \tau in the softmax:
$\sigma_t(z; \tau) = \frac{e^{z_t/\tau}}{\sum_{j \in T} e^{z_j/\tau}}$, and analyzing its effect on convergence and performance could provide deeper insights.

**Response: Thank you for the feedback. We are currently running experiments with the added temperature parameter and will follow up once completed.**

The paper handles continuous treatments by discretizing the treatment space. However, there is no rigorous analysis of the error introduced by this process, such as the approximation error:
Error $= | y(x_i, t) - y(x_i, \tilde{t}) |, where ( \tilde{t} )$ is the discretized treatment. A formal discussion on how the discretization impacts overall performance is needed.

**Response: Thank you for the thoughts on discretization. We are happy to discuss discretization more formally, but we believe our current work stays faithful to real-world applications and therefore how performance can look in the real-world. For example, real-world applications will usually not have too many possible values; medications are only available in certain dose sizes, or pricing of goods is typically done at certain values. We see this in our real-world data, where the raw diabetes dataset only contains 12 possible combinations of the “continuous” prescriptions. Similarly, the raw groceries dataset prices, though theoretically continuous, practically take on only 10 possible values: 1.99, 2.0, 2.49, 2.5, 2.99, 3.49, 3.34, 3.5, 3.99, 4.99. We simplify these to 6 values which are quite close to the original values, 2, 2.5, 3, 3.5, 4, 4.5, 5, as an example in the paper. Finally, we want to note that with our current architecture, it is possible to customize and have more discretized prescriptions.**

The decision to use PCA for dimensionality reduction on high-dimensional embeddings may not be optimal, as it is linear in nature. Nonlinear methods (e.g., autoencoders) might capture the inherent complexity of unstructured data better, which the authors could consider.

**Response: We are currently running our experiments with embeddings extracted from a feedforward neural network trained on our data and will follow up once completed.**

Although the empirical results are promisingCounterfactual Estimation Assumptions, the paper lacks a detailed discussion on the scalability of the proposed method, particularly in high-dimensional and multimodal settings. An analysis of computational complexity would benefit readers assessing its practical deployment.

**Response: Thank you for identifying the scalability of the method. We believe that this is addressed via our experiments, where the size of the groceries and diabetes datasets are larger with 97,295 and 58,200 rows, respectively, compared to the size of the Liver injury dataset with 722 rows. Similarly, we show the method’s efficacy in higher dimensions on both tabular datasets with 54 features and on multimodal datasets with 333 features.**

While the OCT approach provides interpretability, there is no formal quantification of the trade-off between interpretability and performance. A more detailed analysis of the approximation error between the PNN and OCT predictions would be valuable.
**Response: Agreed, this is important. We quantify the approximation error between the PNN and OCT predictions in the form of the OCT’s “training accuracy”, which means, how accurately did the mirrored OCT replicate the prescriptions of the PNN. We have made this clearer in the paper. In Section 3.5, page 9, we have added the following: "We also quantify the approximation error between the PNN and OCT prediction using what we call the “training accuracy” of the OCTs on the prescriptions, which we present in Table 2. For example, a training accuracy of 79\% means that in the training data split, the OCT correctly classified 79\% of the PNN's prescriptions."
In Section 3.6, page 9, we note that Table 3 includes training accuracy, which is defined in the previous section.**

---

### Author Response · Authors · 2025-03-30
**Response to feedback from Reviewer NX5V (Part 1)**

# Response to Reviewer NX5V

Thank you very much for your thorough feedback and thoughtful comments. We have included responses to all of your points, and have also noted the areas that we are currently still working on. We are expecting to have finalized the edits as soon as possible, this coming week.

### Methodological Questions

- Counterfactual Estimation Assumptions: How does treatment assignment bias impact the accuracy of counterfactual estimates, even with doubly robust estimation? Have you tested alternative counterfactual estimation methods (e.g., GAN-based counterfactuals or causal deep learning models)?
  - **Response: Thank you for raising this important point about counterfactual estimation and treatment assignment bias. We fully agree that treatment assignment bias can impact counterfactual estimates — particularly when certain treatment groups are underrepresented, leading to limited support for accurate modeling. While doubly robust (DR) estimation offers protection against misspecification of either the outcome or treatment model, its performance still depends on assumptions like positivity and correct model specification. To address this rigorously, we are now working on including a detailed analysis of the positivity assumption, reporting both effective treatment sample sizes and overlap metrics across treatment groups. We also assess the impact of DR estimation by measuring Standardized Mean Differences (SMDs) before and after reweighting, across all datasets.**
  - **We considered alternative counterfactual methods such as prognostic score matching, but found that they introduce complexity in multitreatment settings. GAN-based models are more data-hungry and could be benchmarked in future works as alternatives.**
- Model Performance and Generalization:
  - How does PNN performance vary with dataset size, particularly for smaller datasets where deep learning models may struggle with overfitting?
    - **Response: Thank you for the question. To address this exact issue, we utilized datasets of various sizes, and we observed consistent results. For example, the diabetes management dataset consists of more than 55,000 rows, whereas the TAVR and Liver Injury datasets are much smaller, with roughly 2,000 and 700 data points each. For smaller datasets, by adjusting the size of the network, the number of parameters that need to be learned is reduced and the training is successful.**
 - Have you evaluated PNNs on external datasets beyond those presented to assess generalizability?
   - **Response: Thank you for the question. The train-test splits we have used aim to address the generalizability concern, since we evaluate our models on unseen data. Overall, it is hard to find high-quality prescriptive datasets, and we wanted to emphasize the applicability of our approach in real-life settings, so we focused on the 6 we present.**
 - How do PNNs perform under distribution shifts or adversarial conditions (e.g., missing data, biased treatment distributions)?
   - **Response: Thank you for the comment. Since Neural Networks cannot be trained using missing values, like e.g. tree methods, we performed KNN imputation. Some of the datasets were complete, while others had more missing values (e.g. TAVR dataset). We also examined the case of biased treatment distributions; the spleen dataset has 35,954 samples overall, and 29,789 belong to the observation class. Under this condition and for this dataset, we observe that the PNNs perform better than the other methods.**
- What are the computational requirements of training and deploying PNNs on large-scale healthcare datasets? How do training times and resource consumption compare with tree-based methods, especially for real-time applications?
   - **Response: Thank you for identifying the scalability of the method. We believe that the computational requirements are reasonable for real-time applications. This is addressed via our experiments, where the size of the groceries and diabetes datasets are larger with 97,295 and 58,200 rows, respectively, compared to the size of the Liver injury dataset with 722 rows. Similarly, we show the method’s efficacy in higher dimensions on both tabular datasets with 54 features and on multimodal datasets with 333 features. Additionally, with tree-based methods, we found scalability to be more difficult, particularly with increasing number of features.**

---

### Author Response · Authors · 2025-03-30
**Response to feedback from Reviewer NX5V (Part 2)**

## Theoretical and Interpretability Questions

Theoretical Guarantees: Can you provide formal proofs or theoretical bounds on the optimality of PNN-prescribed treatments? Under what conditions does the softmax-based treatment assignment converge to an optimal policy?

**Response: Thank you for the comment. We have updated Section 2.4.1 in the paper, and we have included the corresponding part here:**

**The loss function we use shares key properties with the cross-entropy loss function commonly used in multiclass classification problems. The cross-entropy loss is widely used because it is smooth, has bounded gradients, and is Lipschitz continuous. These properties contribute to the efficient convergence of optimization algorithms like SGD and Adam (Kingma & Ba, 2014; Bottou, 2010). Similarly, the loss function employed in this work exhibits these desirable properties. Specifically:**

- **Smoothness: The loss function is smooth because it is a linear combination of the softmax probabilities, which are themselves smooth functions (Bishop & Nasrabadi, 2006).**
- **Bounded Gradients: The gradients of the loss function are bounded because: The weights $Γ_{i,t}$ are bounded due to clipping (as described in Section 2.3.1). The derivative of the softmax function is also bounded (Goodfellow et al., 2016).**
- **Lipschitz Continuity: The loss function is Lipschitz continuous because the softmax function is Lipschitz continuous, and the weights $Γ_{i,t}$ are bounded (Nesterov, 2013).**

**The primary difference between our loss function and the cross-entropy loss is that our loss function uses weights $Γ_{i,t}$ instead of true labels $y_{i,t}$, and it does not include the logarithm of the probabilities. However, these differences do not fundamentally alter the smoothness, boundedness, or Lipschitz continuity of the loss function. As a result, the convergence behavior of our loss function is similar to that of the cross-entropy loss when training feedforward neural networks for multiclass classification problems (LeCun et al., 2015). Under the assumption of bounded weights in the network, a property typically observed when training with SGD or Adam (Ghadimi & Lan, 2013; Reddi et al., 2019), the network will converge to critical points of the loss function. This is consistent with the behavior observed in standard neural network training with cross-entropy loss (Zhang et al., 2016).**

Interpretability and Trustworthiness:
- What strategies can be employed to make PNNs more interpretable beyond Mirrored OCTs?
  - **Response: We agree with you that interpretability is important. We also believe that because interpretability is complex and difficult, and therefore it requires much more research to develop a strong strategy. On that front, we are working to follow up this current paper with another focusing on interpretability of PNNs.**
- How do domain experts (e.g., clinicians) perceive the interpretability of Mirrored OCTs, and do they trust their recommendations?
  - **Response: Application to the real-world is challenging, and we agree that working with domain experts is the next step in actually implementing these prescriptive models and corresponding Mirrored OCT. Though that is a work in-progress, we do have experience implementing prescriptive tree models with healthcare professionals; they have found that trees are easily understood, and they like the clarity of the tree’s splits, both features and the values defining the split. These professionals use these tree models like a third-party opinion on their work; rather than the tree dictating what the professionals do, the tree serves to guide their work or flag any patients that professionals may need to take a second look at.**

---

### Author Response · Authors · 2025-03-30
**Response to feedback from Reviewer NX5V (Part 3)**

### Real-World Deployment and Ethical Considerations

Clinical and Practical Implementation:
What challenges exist in integrating PNNs into clinical decision-making workflows? How would the model handle treatment constraints (e.g., unavailable drugs, insurance limitations, ethical concerns)? How do PNN prescriptions compare to human expert decisions in clinical practice?

**Response: Thank you for bringing up these considerations. We believe that the challenges that PNNs experience are largely similar for many predictive and prescriptive models deployed in real-world clinical workflows. In particular, one major challenge is the trustworthiness of models in clinicians’ eyes. Though we have not yet had an opportunity to deploy our model in a real-world setting, we do have prior experience implementing predictive tree models with healthcare professionals; we have worked closely with the clinicians during the model development in order to build the clinicians’ trustworthiness towards the model. During this development process, clinicians have an increased understanding of the model that they would use once deployed. Then, in practice, these professionals use the model as a third-party opinion on their work; rather than the model dictating what the professionals do, it serves to guide their work or flag any patients that professionals may need to take a second look at. This would hold true with deploying PNNs; if the PNN prescription is the same as the human expert, then it reinforces the clinician’s decision.**

**Alternatively, if the PNN prescription differs from the clinician, then this helps raise a red flag to the clinician to take a closer look at what they are prescribing and why. The clinician takes the responsibility of choosing to keep or modify their prescription.**
**Additionally, to highlight your questions further in our paper, we have included a paragraph on the ethics and limitations of using machine learning models for making decisions in Section 4.4.**
**Finally, regarding the treatment constraints, one can add different penalty terms in the loss function that account for different treatment constraints, drug assignment limitations etc. We added this in the paper too. **
**Regarding the treatment constraints, one can add different penalty terms in the loss function that account for different treatment constraints, drug assignment limitations etc. We added this in the paper too.**

---

### Author Response · Authors · 2025-03-30
**Response to feedback from Reviewer 7LhU (Part 1)**

# Response to Reviewer 7LhU

Thank you very much for your thorough feedback and thoughtful comments. We have included responses to all of your points, and have also noted the areas that we are currently still working on. We are expecting to have finalized the edits as soon as possible, this coming week.

### Lack of causal inference foundations
My main concern is that the paper does not reference foundational causal inference theory. There is no graphical model, no explicit assumptions about confounders, and no discussion on treatment overlap (e.g., is positivity guaranteed?). The paper would benefit significantly from integrating insights from Rubin’s Potential Outcomes Framework or Pearl’s Structural Causal Models, which provide well-established tools for handling confounding bias and ensuring validity of counterfactual estimates.

**Response: Thank you for raising this important point about the causal inference background. We fully agree  that clearly articulating the underlying assumptions—such as unconfoundedness, consistency, and positivity—is essential for the validity of counterfactual estimation. While doubly robust (DR) estimation offers protection against misspecification of either the outcome or treatment model, its performance still depends on assumptions like positivity and correct model specification. To address this rigorously, we are now working on including a detailed analysis of the positivity assumption, reporting both effective treatment sample sizes and overlap metrics across treatment groups. We also assess the impact of DR estimation by measuring Standardized Mean Differences (SMDs) before and after reweighting, across all datasets. These additions are intended to clarify and support the assumptions underlying our counterfactual estimates, and we appreciate the suggestion to make them more explicit.**

### Presentation and organization
Certain sections include excessive details that may not add value for an ML journal audience. Examples include:
- Section 2.4, which provides a tutorial-style explanation of multinomial logistic regression—this could be shortened or omitted.
- Figure 1, which illustrates a generic MLP architecture, does not add significant insight and could be removed. Moreover, Quiza and Davim (2011) should not be the canonical reference for MLPs.
- Section 3.2 (Hyperparameters Tuning), which describes standard neural network hyperparameters in detail, would be more appropriate in an appendix.

**Response: Thank you for these pieces of feedback. For Section 2.4, we have omitted the lengthy discussion of the feedforward neural network and shortened the explanation for the multinomial logistic regression. We have also removed the MLP architecture and updated the MLP reference to Rosenblatt (1958); Rumelhart et al. (1986). In Section 3.2, we have moved the hyperparameters to the appendix.**

### Additional observations and questions
- 1.1 Regress & Compare: This section is missing an appropriate reference.
  - **Response: Thank you for your feedback. We added a reference to Zhao et al. (2012), who introduces a framework that aims to estimate individual treatment rules. The goal is to assign treatments that maximize (or minimize) the expected outcome for each individual, by estimating the potential outcomes under each treatment (using an outcome weighted learning approach) and selecting the treatment that leads to the best outcome.**
- 2.2 “Numerical features are normalized”—Could the authors clarify how the normalization is performed?
  - **Response: Good point. We have clarified what we meant by normalization in Section 2.2: “Numerical features are normalized to the interval [0,1] by subtracting the minimum feature value and dividing by the feature range; we do this to increase stability and equal weighting of features during counterfactual estimation and model.”**
- Table formatting: Some tables contain a lot of content but could be streamlined to focus on key results.
  - **Response: We have moved one of the tables to the Appendix to streamline the focus on key results. We are happy to take further suggestions on how to refine the tables.**

---

### Author Response · Authors · 2025-03-30
**Response to feedback from Reviewer 7LhU (Part 2)**

### Methodological concerns

- Evaluation metrics:
  - Tables 1 and 4 present results using standard deviation instead of standard errors or confidence intervals (CIs). Since overlapping confidence intervals suggest non-significant differences, it would be better to report standard errors and explicitly indicate statistically significant improvements.
    - **Response: Thank you for your feedback. We are in the process of updating the results using standard errors instead of standard deviation.**
  - The number of experiments appears to be ( N = 5 )—could the authors clarify whether this is sufficient for robust statistical comparison?
    - **Response: Thank you for your comment. We have reported our results across 6 different datasets, and for each dataset we performed 5 random train-test splits. This approach allows us to evaluate the stability and robustness of our algorithm across different data distributions and ensure that the results are not overly dependent on a specific split. The random splits ensure that the results are not biased by a particular partitioning of the data, and the performance trends are consistent, which suggests that the results are robust and not due to random variation.**
  - If the performance differences overlap significantly, it would be helpful to highlight methods that clearly outperform others.
    - **Response: Thank you for the comment. Currently, the method that has the best performance in each dataset is highlighted.**
- Missing baselines:
  - DragonNet, TARNet, T-learners, X-learners, S-learners (Kunzel et al. (2019), "Metalearners for Estimating Heterogeneous Treatment Effects."). Please either discuss why they might be relevant or compare against them.
    - **Response: Thank you for the feedback. We are currently looking into their relevancy and preparing to include them as references and possibly as comparison.**
- Data splitting strategy:
  - The paper justifies a 50-50 train-test split in the name of fairness, but the definition of fairness in this context is unclear.
    - **Response: Thank you for pointing this out. Based on this feedback, we have provided a more detailed justification for the 50-50 train-test split in Section 3.1 and have omitted the use of the term “fairness” to avoid confusion in definition.**
  - A cross-validation approach would likely provide more robust and less biased performance estimates. Could the authors comment on why it was not used?
    - **Response: Thank you for the comment. The multiple random train-validation-test splits were selected since in a real-life application, it is more likely for someone to just split their dataset in train-validation-test sets and then select the best model for deployment on the validation set. To imitate this scenario, we wanted to examine the robustness of the results if different splits were selected. Also, the 50-50 split that we want cannot be achieved through cross validation.**
- Dimensionality reduction (PCA):
  - The authors state that PCA is optionally applied to improve stability and performance. Could this claim be validated with an ablation study to assess whether PCA meaningfully improves results?
    - **Response: We are currently running our experiments with embeddings extracted from a feedforward neural network trained on our data and will follow up once completed. We are also including the raw embeddings extracted from the pretrained model, to study the effect of dimensionality reduction.**
- Table 3 (Training Accuracy):
  - The purpose of reporting training accuracy is unclear. If its inclusion is necessary, it should be moved to the appendix—or, if not particularly insightful, removed altogether.
    - **Response: We quantify the approximation error between the PNN and OCT predictions in the form of the OCT’s “training accuracy”, which means, how accurately did the mirrored OCT replicate the prescriptions of the PNN. We have made this clearer in the paper. In Section 3.5, page 9, we have added the following: "We also quantify the approximation error between the PNN and OCT prediction using what we call the “training accuracy” of the OCTs on the prescriptions, which we present in Table 2. For example, a training accuracy of 79\% means that in the training data split, the OCT correctly classified 79\% of the PNN's prescriptions." In Section 3.6, page 9, we note that Table 3 includes training accuracy, which is defined in the previous section.**

---

### Author Response · Authors · 2025-04-09
**Response to remaining comments from Reviewer 1nqP and NX5V**

Please see our responses to the remaining comments from all reviewers. We have updated the paper in the submission as well. We look forward to hearing back!

# Response to Reviewer 1nqP

Sensitivity to Propensity Scores: In the doubly robust estimator, if the propensity scores $p_{i,t}$ are very small, the term $\frac{1}{p_{i,t}}$ may induce high variance. The authors should discuss potential safeguards, such as clipping $p_{i,t}$, to stabilize the estimates.

**Response: Thank you for the feedback. To reduce the potential instability that arises when we divide with the propensity score, we clip the ones that are smaller than a certain value \citep{lee2011weight}. We generally choose a clipping threshold of 0.01-0.05, depending on the resulting rewards' values. We reran our experiments using the newly calculated rewards.**


While the softmax function is used to approximate the non-differentiable indicator function, the impact of the smoothing parameter is not explored. Introducing a temperature parameter \tau in the softmax:
$\sigma_t(z; \tau) = \frac{e^{z_t/\tau}}{\sum_{j \in T} e^{z_j/\tau}}$, and analyzing its effect on convergence and performance could provide deeper insights.

**Response: Thank you for the feedback. We have added a Section on convergence properties of our loss function (Section 2.4.1).**

The decision to use PCA for dimensionality reduction on high-dimensional embeddings may not be optimal, as it is linear in nature. Nonlinear methods (e.g., autoencoders) might capture the inherent complexity of unstructured data better, which the authors could consider.

**Response: We have run extra experiments with embeddings extracted from a feedforward neural network trained on our data, as well as PCA-reduced embeddings and full-dimensional embeddings. Across both datasets, dimensionality reduction leads to improved performance compared to using full-dimensional embeddings. For TAVR, representations extracted from the classification head yield the best results, suggesting that supervised fine-tuning can produce more informative and stable embeddings. In the liver trauma dataset, both classification head and PCA-reduced embeddings outperform the full-dimensional version, with PCA achieving the highest improvement. These results suggest that dimensionality reduction can improve model stability and performance, especially when the original embedding dimension is high relative to dataset size. For more details please refer to Appendix Section A1.**

# Response to Reviewer NX5V

### Methodological Questions

Counterfactual Estimation Assumptions: How does treatment assignment bias impact the accuracy of counterfactual estimates, even with doubly robust estimation? Have you tested alternative counterfactual estimation methods (e.g., GAN-based counterfactuals or causal deep learning models)?

**Response: Thank you for raising this important point about the causal inference background. We fully agree  that clearly articulating the underlying assumptions—such as unconfoundedness, consistency, and positivity—is essential for the validity of counterfactual estimation. While doubly robust (DR) estimation offers protection against misspecification of either the outcome or treatment model, its performance still depends on assumptions like positivity and correct model specification. To address this rigorously, we have included a detailed analysis of the positivity assumption (Section 4.1), reporting both effective treatment sample sizes and overlap metrics across treatment groups. We also assess the impact of DR estimation by measuring Standardized Mean Differences (SMDs) before and after reweighting, across all datasets. These additions are intended to clarify and support the assumptions underlying our counterfactual estimates, and we appreciate the suggestion to make them more explicit.**

---

### Author Response · Authors · 2025-04-09
**Response to remaining comments from Reviewer 7LhU**

Please see our responses to the remaining comments from all reviewers. We have updated the paper in the submission as well. We look forward to hearing back!

# Response to Reviewer 7LhU

### Lack of causal inference foundations
My main concern is that the paper does not reference foundational causal inference theory. There is no graphical model, no explicit assumptions about confounders, and no discussion on treatment overlap (e.g., is positivity guaranteed?). The paper would benefit significantly from integrating insights from Rubin’s Potential Outcomes Framework or Pearl’s Structural Causal Models, which provide well-established tools for handling confounding bias and ensuring validity of counterfactual estimates.

**Response: Thank you for raising this important point about the causal inference background. We fully agree  that clearly articulating the underlying assumptions—such as unconfoundedness, consistency, and positivity—is essential for the validity of counterfactual estimation. While doubly robust (DR) estimation offers protection against misspecification of either the outcome or treatment model, its performance still depends on assumptions like positivity and correct model specification. To address this rigorously, we have included a detailed analysis of the positivity assumption (Section 4.1), reporting both effective treatment sample sizes and overlap metrics across treatment groups. We also assess the impact of DR estimation by measuring Standardized Mean Differences (SMDs) before and after reweighting, across all datasets. These additions are intended to clarify and support the assumptions underlying our counterfactual estimates, and we appreciate the suggestion to make them more explicit.**

### Evaluation metrics:
Tables 1 and 4 present results using standard deviation instead of standard errors or confidence intervals (CIs). Since overlapping confidence intervals suggest non-significant differences, it would be better to report standard errors and explicitly indicate statistically significant improvements.

**Response: Thank you for your feedback. We have updated the results using standard errors instead of standard deviation.**

Missing baselines: DragonNet, TARNet, T-learners, X-learners, S-learners (Kunzel et al. (2019), "Metalearners for Estimating Heterogeneous Treatment Effects."). Please either discuss why they might be relevant or compare against them.

**Response: Thank you for the feedback. We have included them and a discussion of their relevance as related work in the related literature section.**

Dimensionality reduction (PCA): The authors state that PCA is optionally applied to improve stability and performance. Could this claim be validated with an ablation study to assess whether PCA meaningfully improves results?

**Response: We have run extra experiments with embeddings extracted from a feedforward neural network trained on our data, as well as PCA-reduced embeddings and full-dimensional embeddings. Across both datasets, dimensionality reduction leads to improved performance compared to using full-dimensional embeddings. For TAVR, representations extracted from the classification head yield the best results, suggesting that supervised fine-tuning can produce more informative and stable embeddings. In the liver trauma dataset, both classification head and PCA-reduced embeddings outperform the full-dimensional version, with PCA achieving the highest improvement. These results suggest that dimensionality reduction can improve model stability and performance, especially when the original embedding dimension is high relative to dataset size. For more details please refer to Appendix Section A1.**

---

### Decision · Action_Editor_w5tx · 2025-04-20

**Recommendation:** Reject

**Comment:**

The paper introduces Prescriptive Neural Networks (PNNs), a deep learning framework designed to prescribe optimal treatments by leveraging multimodal data which is omnipresent, especially in real-world scenarios. The paper moves from classification to prescription and demonstrates strong performance on different treatment scenarios. The idea is very interesting and has applications on several real-world scenarios but the reviewers still has several concerns after the rebuttal. The major points of contention were as follows:

1. Reviewer 7LhU mentions that the best performing methods lack the statistical significance to be claimed as such (e.g. 1.83-+0.23 is not better than 1.73-+0.21). This is a methodological issue that the reviewer had raised and has not been yet resolved.

2. Reviewer 7LhU also points out the issue with 50%-50% splits as tghis is not very common and if the lack of data justifies the use of such dramatic split, at least the authors should attempt to do bootstraping and provide a more robust estimate (what if the performance is very sensitive to different splits?) This is not captured anywhere at the moment.

3. Reviewer NX5V points out that there were initial hesitations around the causal foundations and its proper explorations in the manuscript. ALthough some viable modifications have been proposed, the paper still requires major revisions before acceptance.

Overall, I do think that the paper has a major potential and if all the above concerns are addressed then it will be fit for publication. I recommend rejection as of now but strongly encourage the authors to implement the changes and re-submit.

**Audience:**

Yes. A broad set of audience will be interested in this work.

**Claims And Evidence:**

No. The claims in the submission seem accurate but the overall evidence to support the claims are not very well fleshed out according to the reviewers.

**Resubmission Of Major Revision:**

The authors may consider submitting a major revision at a later time.